# AutoRL Hyperparameter Landscapes

Aditya Mohan[*1]  Carolin Benjamins[*1]  Konrad Wienecke[1]  Alexander Dockhorn[2]
Marius Lindauer[1]

[1]Institute of Artificial Intelligence, Leibniz University Hannover, Germany
[2]Institute for Information Processing, Leibniz University Hannover, Germany

**Abstract**  Although Reinforcement Learning (RL) has shown to be capable of producing impressive results, its use is limited by the impact of its hyperparameters on performance. This often makes it difficult to achieve good results in practice. Automated RL (AutoRL) addresses this difficulty, yet little is known about the dynamics of the *hyperparameter landscapes* which hyperparameter optimization (HPO) methods traverse in search of optimal configurations. In view of existing AutoRL approaches dynamically adjusting hyperparameter configurations, we propose an approach to build and analyze these hyperparameter landscapes not just for one point in time but at multiple points in time throughout training. Addressing an important open question on the legitimacy of such dynamic AutoRL approaches, we provide thorough empirical evidence that the hyperparameter landscapes strongly vary over time across representative algorithms from RL literature (DQN, PPO, and SAC) in different kinds of environments (Cartpole, Bipedal Walker and Hopper). This supports the theory that hyperparameters should be dynamically adjusted during training and shows the potential for more insights on AutoRL problems that can be gained through landscape analyses. Our code can be found at `https://github.com/automl/AutoRL-Landscape`

## 1 Introduction

The combination of RL techniques with the power of function approximation inherent in Deep Learning has led to several impressive successes (Silver et al., 2016; Zhou et al., 2017; Bellemare et al., 2020; Badia et al., 2020; Lee et al., 2020; Degrave et al., 2022). As research in RL soars and the field targets increasingly harder learning-based optimization and control problems, extracting good performance out of ever more complicated pipelines becomes the need of the hour. Thus, techniques in Automated Reinforcement Learning (AutoRL; Parker-Holder et al. (2022)) have started to automate the design of RL approaches.

One goal of AutoRL is hyperparameter optimization (HPO), whereby AutoRL determines hyperparameter configurations that can help an RL agent achieve the best performance. However, the distribution shift induced by the RL agent generating its own learning data via interactions with the environment leads to non-stationarity in the learning process. Consequently, RL pipelines can be very sensitive to hyperparameter configuration (Henderson et al., 2018; Parker-Holder et al., 2022), making it difficult to find an optimal static configuration at the beginning of the training. Thus, Parker-Holder et al. (2022) argue for the necessity to adjust hyperparameters throughout the training process in RL. Although several AutoRL approaches (Li et al., 2019; Parker-Holder et al., 2020; Dalibard and Jaderberg, 2021; Wan et al., 2022) try to exploit this property, to date, there is no thorough study validating this hypothesis. In our search for better AutoRL methods, we provide insightful evidence of how the hyperparameter landscape changes throughout time. To this end, we propose a structured approach to collect performance data per time and landscape analysis methods.

**Contributions**. (i) We introduce a pipeline for creating hyperparameter landscapes of dynamic configurations at multiple discrete time steps throughout training (see Figure 1). (ii) We delineate

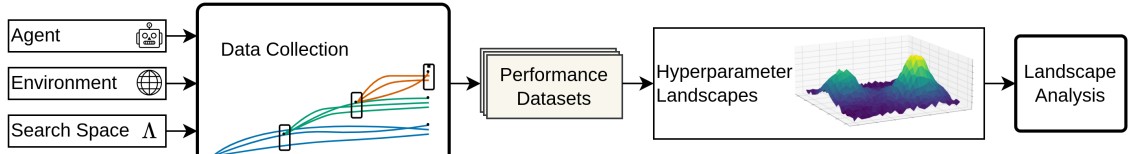

Figure 1: An overview of our hyperparameter landscape creation and analysis pipeline. With an RL algorithm, environment, and the hyperparameter search space, we collect performance data for hyperparameters covering the search space *at multiple time steps* throughout training (Section 4.1). The gathered data relates algorithm performance to the algorithm configuration, which we use for modeling the landscapes (Section 4.2).

methods with which the landscapes can be inspected for traits such as their general structure, configuration stability, and hyperparameter importance. (iii) We demonstrate the insights generated by our extensive study pipeline with DQN (Silver et al., 2016), PPO (Schulman et al., 2017) and SAC (Haarnoja et al., 2018) on Cartpole, Bipedal Walker and Hopper (Brockman et al., 2016a) environments.

## 2 Related Work

To the best of our knowledge, our work is the first to address and inspect hyperparameter landscapes in RL. Our study also sheds light on the properties of hyperparameter values that are specific to the RL pipeline.

**Automated Reinforcement Learning (AutoRL)**. The goal of AutoRL is to facilitate deploying well-performing RL pipelines by making the process of designing RL algorithms data-driven. Parker-Holder et al. (2022) categorize AutoRL approaches on the basis of four major design decisions: task design, algorithm selection, the architecture of the policy network, and hyperparameters. Our work specifically focuses on analyzing the impact of hyperparameters across different types of algorithms on environments with different dynamics. Islam et al. (2017) analyze multiple *static* hyperparameters of two RL algorithms and two environments, noting that differences exist over the different RL contexts. Shala et al. (2022) created a tabular benchmark to compare the reward curves of well-established RL methods across multiple environments and hyperparameter configurations. Our work adds principle to this process by visualizing hyperparameter landscapes at different points in time.

**Landscape Analyses**. Landscape analyses have traditionally been a part of the optimization literature (Pitzer and Affenzeller, 2012) where the quality of different search solutions is measured using a fitness function. In HPO, hyperparameter landscapes are closely related to a given performance metric (e.g., the *validation loss* of a neural net in supervised learning, or the *evaluation return* in RL) by mapping hyperparameter configurations to the performance metric. Landscapes additionally require a notion of a neighborhood or distance to be able to relate and interpolate between different hyperparameter configurations (Stadler, 2002).

Through their structured view of the model's performance, hyperparameter landscapes provide a perspective on the central subject of HPO, and analysis can reveal how to search for optima efficiently. Pimenta et al. (2020) analyze hyperparameter landscapes for highly nested search spaces. Pushak and Hoos (2022) show that AutoML loss landscapes are often much more structured than assumed, allowing for cheap, independent optimization of the hyperparameters. In algorithm configuration, Pushak and Hoos (2020) showed that benign characteristics of configuration landscapes (Pushak and Hoos, 2018) can be exploited for efficient optimizers. Further, Malan (2021) provide an overview of a wide range of landscape analysis techniques.

**Dynamic Configurations.** While dynamic configurations can already be advantageous for stationary problems (Jaderberg et al., 2017; Chen et al., 2023), the non-stationary of RL can give them an even bigger edge over static configurations (Li et al., 2019; Parker-Holder et al., 2020; Dalibard and Jaderberg, 2021; Wan et al., 2022; Parker-Holder et al., 2022). Adriaensen et al. (2022) expands on the use of both RL and other optimization techniques for inferring configuration schedules, showing that these schedules can outperform static configurations for algorithms from multiple artificial intelligence disciplines (though not including RL). RL itself can also be used to find optimized configuration schedules (Biedenkapp et al., 2020). Our work adds to this line of work by providing insights into the impact of dynamic configurations of RL hyperparameters.

## 3 Preliminaries

In the following, we summarize the main background necessary for our approach to studying the properties of AutoRL landscapes.

### 3.1 Reinforcement Learning

Reinforcement Learning (RL) deals with sequential decisions making problems, where an *agent* interacts with an *environment*. One way to model such scenarios is by using a Markov Decision Process (MDP), represented as a 5-tuple $\mathcal{M} = \langle \mathcal{S}, \mathcal{A}, P, R, \rho \rangle$.

The environment is in some *state* $s \in \mathcal{S}$. The agent takes an *action* $a \in \mathcal{A}$ that results in a *transition* of the environment from the current state $s$ to the *next state* $s' \in \mathcal{S}$. The transition function $P : \mathcal{S} \times \mathcal{A} \to \Delta(\mathcal{S})$ governs this transition by taking a state $s$ and action $a$ as inputs and outputting a probability distribution $\Delta(\cdot)$ over the next states, from which $s'$ can be sampled. For each transition, the agent receives a *reward* according to a reward function $R : \mathcal{S} \times \mathcal{A} \times \mathcal{S} \to \mathbb{R}$. Each of these sequences, represented as the tuple $(s, a, r, s')$, is also referred to as an experience. The initial state $s_0$ is sampled from the distribution $\rho$.

The agent selects actions using a policy $\pi : \mathcal{S} \to \Delta(\mathcal{A})$ that produces a probability distribution over actions given a state. This definition also encompasses deterministic policies that output a single action given a state by using a delta distribution. At each timestep, the agent acts according to its policy $\pi$ to generate a *trajectory* of experiences $\tau = (s_0, a_1, r_1, \ldots, s_T)$ for a *horizon* $T$. In this work, we focus on episodic settings where the returns are accumulated till the end of episodes before the optimization is performed. Additionally, we use the common practice of discounting the returns subsequent to the starting state with a factor $\gamma \in [0, 1]$ (Discounted RL; Dewanto and Gallagher (2022)). The expected sum of these rewards is called a return

$$G(\pi, s) = \mathbb{E}_{(s_0=s, a_1, r_1, \ldots, s_T) \sim \pi} \left[ \sum_{t=1}^{T-1} \gamma^{t-1} r_t \right]. \tag{1}$$

The agent's objective is to learn an optimal policy $\pi^* \in \Pi$ that maximizes $G(s)$

$$\pi^* \in \underset{\pi \in \Pi}{\operatorname{argmax}} \, \mathbb{E}_{s_0 \sim \rho} \left[ G(\pi, s_0) \right]. \tag{2}$$

It is important to note that our approach does not depend on the setting being episodic and discounted and can be extended to Continual (Khetarpal et al., 2022) and Average Reward (Dewanto et al., 2020) settings. However, we leave such analyses to future work.

### 3.2 The Learning Process

In Deep RL, a policy is a Deep Neural Network parameterized by $\theta \in \mathbb{R}^n$. Improving or learning the policy entails rolling out the current policy $\pi_\theta$ for a number of steps on an MDP $\mathcal{M}$, and collecting the experiences in a trajectory $\tau$. Using the collected experiences, the policy is improved by minimizing an appropriate objective $J(\theta)$, which either reflects a form of TD-Learning (Sutton,

1988) or utilizes the Policy Gradient (Sutton et al., 1999). We consider $\mathcal{J}$ to be the set of possible objectives.

In addition to $\mathcal{M}$, learning also depends on the seed $d \in \mathbb{N}$ controlling the initial state distribution $\rho$ and the randomness within the policy improvement procedure, as well as a set of hyperparameters $\boldsymbol{\lambda} \in \boldsymbol{\Lambda}$ that control the learning algorithm. This usually includes quantities like the discount factor $\gamma$ or the learning rate $\alpha$.

We begin the characterization of the learning process by subsuming all the factors that affect learning into the notion of an algorithm. Thus, an algorithm $Z$ takes all of these as input and produces a new set of weights $\boldsymbol{\theta}' = Z(\boldsymbol{\theta}, \mathcal{M}, d, \boldsymbol{\lambda}, J(\boldsymbol{\theta}))$. With a slight abuse of notation, we can subsume $\boldsymbol{\theta}$ into the policy definition since its usage is tightly coupled with the policy. Additionally, we do not consider settings involving Transfer Learning (Zhu et al., 2020) and Generalization (Kirk et al., 2023) in this work, leading to the MDPs in our case differing only in the initial state distribution conditioned on the seed $\rho(d)$, and the transition operator $P(d)$. These are already included by explicitly conditioning $Z$ on $d$. Consequently, we can remove $\mathcal{M}$ from this definition as well, allowing us to rewrite the algorithm definition as the mapping

$$Z : \Pi \times \mathcal{N} \times \boldsymbol{\Lambda} \times \mathcal{J} \to \Pi \qquad \pi' = Z(\pi, d, \boldsymbol{\lambda}, J) \tag{3}$$

We characterize the performance of $Z$ by looking at the distribution of (undiscounted) *evaluation returns* $\mathbb{E}_{s_0 \sim \rho} G(\pi_Z, s_0)$ of policy $\pi_Z$ obtained by $Z$ from a starting state $s_0$. In practice, we can only approximate this distribution through either modeling or sampling.

## 3.3 Fitness Landscapes

Fitness functions guide the optimization process to solve an objective by measuring the quality of solutions being generated. Given a fitness measure, potential solutions can be compared based on their values measured by the fitness function. The fitness function can further be extended into a fitness landscape by introducing some form of topology onto the search space.

Malan (2021) define a landscape on the basis of three elements: (i) A set $X$ of configurations (i.e., solutions to the problem). (ii) A notion of neighborhood, nearness, distance, or accessibility on $X$. (iii) A fitness function $f : X \to \mathbb{R}$ that maps this configurations to a fitness value.

The notion of the algorithm introduced in Equation (3) can now be used to create a landscape by considering $X$ to be the set of hyperparameter values $\boldsymbol{\lambda}$, and $G(\pi_Z, s_0)$ - policy obtained from applying $Z$ to a starting state $s_0$ - to be the fitness of the algorithm that depends on this configuration. Hence, by sampling multiple hyperparameter configurations and measuring the fitness of the algorithms that use these configurations, we can create a landscape by plotting the topology resulting from aggregating the return distributions.

## 4 Method

We propose a systematic approach for studying RL hyperparameter landscapes with two objectives: (i) what are the properties of these landscapes (e.g., convexity or modality), following ideas from AutoML landscapes (Pushak and Hoos, 2022), and (ii) how do these landscapes change during the training process, following the assumption of dynamic configuration (Li et al., 2019; Parker-Holder et al., 2020; Dalibard and Jaderberg, 2021; Wan et al., 2022). In order to efficiently build our RL hyperparameter landscape at different points in time during training, we need a good data collection process and a method to model the landscape. We describe both of these in the following sections.

It is important to note that for collecting the data for the hyperparameter landscapes, we take a greedy approach and try to imitate an optimizer that would always go for the best possible hyperparameter configuration schedule. So, we are not interested in all possible landscape changes but only in those that are relevant for building successful AutoRL approaches.

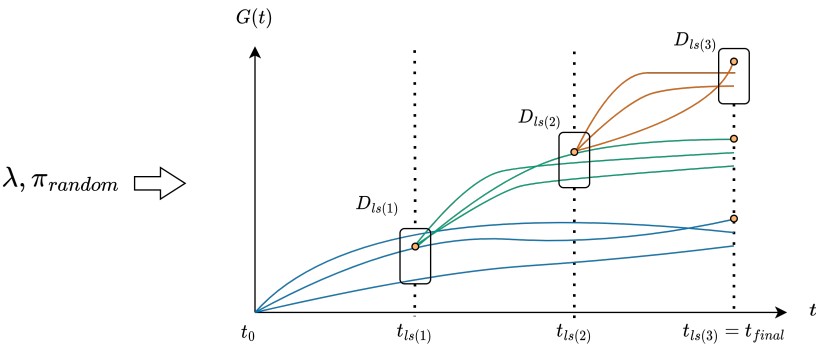

Figure 2: Overview of the data collection process for landscapes with three configurations $\lambda$ and three phases. We initialize the process by training a random RL policy $\pi_{random}$ on each configuration $\lambda \in \lambda$. The three configurations run till the first landscape point $t_{ls(1)}$ which forms the first landscape dataset $D_{ls(1)}$. The policies are snapshotted at this point, and the policy for the next phase is selected based on the final performance, indicated by the continuation of the blue points. The selected configuration is shown with orange circles. This process is repeated for two more phases to create landscape datasets $D_{ls(1)}$ and $D_{ls(3)}$. The end of the final phase $t_{ls(3)}$ corresponds to the final training point $t_{final}$

## 4.1 Data Collection

Figure 2 outlines the overview of our data collection process. Given a training environment $\mathcal{M}$, we divide the learning timeframe into different *phases* $0 < t_{ls(1)} < \cdots < t_{final}$, where $t = t_{ls(i)}$ (*ls* denoting landscape) denotes the time point for collecting landscape data, and $t = t_{final}$ is at the end of training. Each phase entails using a checkpoint from the last phase to initialize algorithms differentiated by their seeds and HP configurations. At the end of the phase, we evaluate the fitness through the returns $\hat{G}_z$. We explain this process further in the following paragraphs.

**Sampling Configurations**. At the start of each phase $i$, we consider a set of hyperparameters $\lambda \in \Lambda$ that can characterize $\Lambda$ by providing sufficient coverage of the areas that we are interested in. We sample values of $\lambda$ using a scrambled Sobol sampling strategy (Sobol, 1967; Joe and Kuo, 2008), which mitigates the inefficiency of grid search and the issue of sufficient search-space coverage of the random search.

**Notion of Distance**. The codomain $[0,1]^n$ of the Sobol sampler additionally acts as a normalized view of the search space, and thus, a distance metric in this space can additionally map to $\Lambda$. We model this using a monotone function $u : [0,1]^n \to \Lambda$.

**Training and Evaluation**. For each sampled configuration $\lambda$, we consider a set of seeds $D$ and instantiate an algorithm for each seed-configuration combination while keeping the objective $J$ constant, thus, resulting in $|D| \times |\lambda|$ algorithms. The input policy for all the algorithms is the best policy from the last phase $\pi_{i-1}^*$.

Each of the instantiated algorithms is then run till the end of the phase, and the returns are collected into a dataset $\mathcal{D}_{ls(i)}$ which signifies the fitness of each algorithm. The returns are computed across all the $|D|$ seeds. Thus, each element of the dataset contains fitness evaluations of a tuple $C = \{\lambda_1, \ldots, \lambda_{|D|}\}$. This gives us a new set of policies of the current phase $\Pi_i \subset \Pi$.

Our next task is to select the best policy $\pi^* \in \Pi_i$. Since early performances do not accurately reflect final performances in RL (Shala et al., 2022), we select policies based on their final performances instead. For this, we train the algorithms till the final timestep $t_{final}$ and then evaluate them. To mitigate the noise in the final evaluation, we aggregate evaluations conducted at $0.95 t_{final}$,

$0.975 t_{final}$, and $t_{final}$ steps into a mean value and use this as a fitness value. This constitutes the dataset $\mathcal{D}_{final(i)}$.

**Snapshots and Configuration Selection**. We save the intermediate policies in each phase as snapshots of the network parameters. We then choose one of these snapshots in the final phase based on $\mathcal{D}_{final(i)}$. To perform this selection, we first choose the configuration set with the highest Interquartile Mean (IQM) and then the initialize configuration by a seed corresponding to the highest IQM in the selected set. IQM as an aggregation mechanism allows us to mitigate the outlier bias prevalent in mean aggregation while incorporating more data in our evaluation than median aggregation (Agarwal et al., 2021). With $\boldsymbol{\lambda^*}$ and $d^*$ selected from this phase, we can train the algorithms in the next phase by providing these and the previous best policy $\pi_{i-1}^*$ to the respective algorithm. The output of the algorithm gives us the best policy for the next stage.

## 4.2 Landscape Modeling and Analysis

We estimate the approximate statistics of the landscapes using $\mathcal{D}_{ls(i)}$ of each phase. Since the performance of different seeds can be very different, just getting the mean and standard deviation of the distribution is not very insightful. Instead, we model three variants of the landscape to take the behavior of dynamic AutoRL approaches into account. We first calculate the mean IQM of the landscape, which describes the typical performance expected from the algorithm. We then calculate the upper and lower quantiles encompassing 95% of the samples[1]. A landscape model then encompasses three functions $f_{\{upper,mean,lower\}} : \Lambda \rightarrow \mathbb{R}$ that use these statistics to map out the hyperparameter landscape over the search space. We call these functions the *upper*, *mean*, and *lower* surfaces. Each surface is independently modeled from either the IQM or the quantiles of the return distribution of each configuration.

**Landscape Models**. Given the notion of distance defined in the codomain of the sampler $[0, 1]^n$, we first map the surfaces to a unit hypercube by $f'_{\{upper,mean,lower\}} : [0, 1]^n \rightarrow [0, 1]$, and then interpret it using the distance transformation $u$.

We use two models of the landscape. The first is **Interpolated Landscape Models (ILM)**. We leverage RBF interpolation with a linear Kernel to construct a continuous surface over the search space from the given samples. This surface meets every input point without any filtering or generalizing being applied. The second model family is **Independent Gaussian Process Regressors (IGPRs)**. Although Gaussian Processes (GPs) can inherently model uncertainty which could be used to fully model lower and upper quantiles as well as the mean of normal distributions, we instead use just the mean of the GP to model the surfaces independently. We use an RBF kernel and optimize parameters and length scales with `scipy`'s `L-BFGS-B` optimizer. Unlike the ILMs, the IGPRs do generalize over the input samples, presenting a different view of the underlying data and, based on our experiments, leading to smoother landscapes that show more global patterns.

**Landscape Analysis**. Inspired by Pushak and Hoos (2022), we use Individual Conditional Expectation (ICE) curves (Goldstein et al., 2015) to model one-dimensional slices through the hyperparameter landscape. Specifically, we create one curve for each choice of the fixed hyperparameters. By showing individual effects, these curves can be used to compare the isolated effects of individual hyperparameters to one another.

**Modality**. Modality of cost distributions in Safe-RL has been shown to be an interesting property (Yang et al., 2023). Unimodal distributions lead to conservative and stable approximations at the cost of expressivity, while multimodal distributions add expressivity at the cost of stability. We visualize the degree to which configurations produce unimodal performance distributions by analyzing

---

[1]Precisely, the mean $k$% of samples are encompassed by the $\left(0 + \frac{100-k}{2}\right)$-quantile and the $\left(100 - \frac{100-k}{2}\right)$-quantile.

the collected performance samples of each configuration. To decide whether a set of samples is unimodal, we employ the folding test of unimodality by Siffer et al. (2018). Intuitively, the test looks at a data distribution and tries to find a pivot point around which the distribution can be folded to reduce the variance. Thus, if a data distribution is multi-modal, then folding will result in a high variance reduction, while this would not be the case for distributions that are unimodal. The test outputs a folding statistic $\Phi$, which is the ratio of the variance after folding to the initial variance. Thus, $\Phi \geq 1$ signifies that the distribution is rather unimodal while $\Phi < 1$ signifies that the distribution is rather multimodal. We filter out results where $p \geq \alpha$ with $\alpha = 0.05$.

## 5 Experiments

We first present a general overview of our experimental setup, and then the hyperparameter landscapes of the phases through visualizations of demonstrative landscape surfaces. We then review our results for per-configuration unimodality. Please refer to Appendix A for full plots.

### 5.1 Experimental Setup

We construct the hyperparameter landscapes for DQN (Mnih et al., 2015) on gym's Cartpole (Brockman et al., 2016b), SAC (Haarnoja et al., 2018) on gym's Hopper-v3, and PPO (Schulman et al., 2017) on Bipedal-Walker-v2 (Brockman et al., 2016b). These combinations ensure (i) diverse environments dynamics, since the two selected environments vary by a great degree in their physical dynamics and convergence requirements; (ii) coverage of both kinds of policy objectives, since DQN uses TD-error while SAC uses policy loss and (iii) diverse exploration strategies, since DQN and SAC follow two very distinct archetypes of exploration strategies in RL (Amin et al., 2021)

We sample 128 configurations and train them with 5 different environment seeds. We additionally use two separate seeds, one for sampling the configurations and the other for evaluating the configurations. Table 1 shows the hyperparameters considered in our landscape analysis for DQN and SAC. We consider three phases for DQN at 50000, 100000, 150000 timesteps, four phases for SAC at 125000, 250000, 375000, 500000, and three phases for PPO (at steps 50000, 100000, 150000). Find our code here: `https://anon-github.automl.cc/r/autorl_landscape-F04D`.

| DQN | | | SAC | | | PPO | | |
|---|---|---|---|---|---|---|---|---|
| HP | Range | Scale | HP | Range | Scale | HP | Range | Scale |
| Learning rate $\alpha$ | $[1e^{-4}, 0.1]$ | Log | Learning rate $\alpha$ | $[1e^{-4}, 0.1]$ | Log | Learning rate $\alpha$ | $[1e^{-4}, 0.1]$ | Log |
| Discount Factor $\gamma$ | $[0.8, 0.9999]$ | Log | Discount Factor $\gamma$ | $[0.8, 0.9999]$ | Log | Discount factor $\gamma$ | $[0.8, 0.9999]$ | Log |
| Final Epsilon $\epsilon_f$ | $[0.01, 1]$ | Linear | Polyak Update $\tau$ | $[1e^{-4}, 0.2]$ | Log | Generalized advantage estimate $\lambda_{gae}$ | $[0.8, 0.9999]$ | Log |

Table 1: Hyperparameters considered as part of $\lambda$ in DQN, SAC and PPO

### 5.2 Landscape Inspection

Figure 3 shows the mean surface of the IGPR plots for DQN, while Figure 4 shows the same for SAC and Figure 5 for PPO. As can be seen, the landscapes strongly vary in their structure over the phases. Thus, they confirm that throughout training, the effect of hyperparameters as well as their optimal settings vary in this experiment. In this sense, the results promote the use of dynamic configurations, setting a precedent for research on other RL contexts. A deeper look at the plots shows that the performance peaks move strongly for different hyperparameters, indicative of both the environment complexity and optimization procedure.

In the case of DQN, we see that the peak occurs in a narrow region for both learning rate $\alpha$ and discount factor $\gamma$ around the final phase. However, the variation is stronger across the range of $\gamma$ while almost negligible in the case of $\alpha$, indicating that $\gamma$ largely influences the scores on its own with peaks around 0.984 in the final phase. For SAC, on the other hand, the behavior is variably different. The peak performance region remains in between $[0.9841, 9.9984]$ for $\gamma$ throughout, while



(a) Phase 1        (b) Phase 2        (c) Phase 3

Figure 3: IGPR plots of the mean surfaces for learning rate and discount factor for DQN across three phases of the RL training process. The local minima are represented by the inverted triangle and the maxima by the normal triangle. The configuration selected for the next stage is represented by a star



(a) Phase 1     (b) Phase 2     (c) Phase 3     (d) Phase 4

Figure 4: IGPR plots for learning rate and discount factor for SAC across phases

around the final phase, we see an increasing number of values for $\alpha$ producing near maximum performance. This indicates that in a significantly more complex environment, SAC is able to explore more efficiently with the right range of $\gamma$ and thus, requires fewer variations in hyperparameter schedules, which corroborates with the advantage of soft updates and entropy-based exploration inherent in SAC. Consequently, potential HP schedules for $\alpha$ and $\gamma$ would have a greater impact on the learning of TD-based off-policy algorithms such as DQN, something that could be potentially attributed to the learning dynamics of TD-algorithms themselves (Lyle et al., 2022).

For PPO, from the IGPR approximations of the mean surface in Figure 5 we see that there is one region with a high performance whose location also changes over the phases. This implies that PPO is less robust to HP decisions in general. In addition, we investigated the model fit with cross-validation and see that the different model types, ILM and IGPR, fit very similarly. They both fit the performance data of PPO and DQN quite well whereas there are higher errors on SAC. For more details see Appendix B.

Overall, these landscapes provide an overview of the way the performance of RL algorithms behaves and these depend very much on the context in which HPO is being applied. The dynamic nature is not just related to the hyperparameter configuration but also to the optimization problem at hand. Thus, HPO should not just be focused on static configurations, but additional properties of the optimization process should be incorporated to discover suitable HPO schedules, where needed.

## 5.3 Per-configuration modality

Figure 6 and Figure 7 show the discretized modality analysis of our collected data, mapped over the search space for learning rate $\alpha$ and discount factor $\gamma$. Additionally, Table 2 presents the overall sizes of the three categories (unimodal, multimodal, and uncategorized). We regard configurations that produce unimodal return distributions to be more stable than those which produce multimodal ones.

Generally, more return distributions are categorized as multimodal rather than unimodal. This is especially true for the last phase, where we find 49.22% of configurations to be multimodal for



(a) Phase 1       (b) Phase 2       (c) Phase 3

Figure 5: IGPR plots of the mean surfaces for learning rate and discount factor for PPO across three phases of the RL training process.

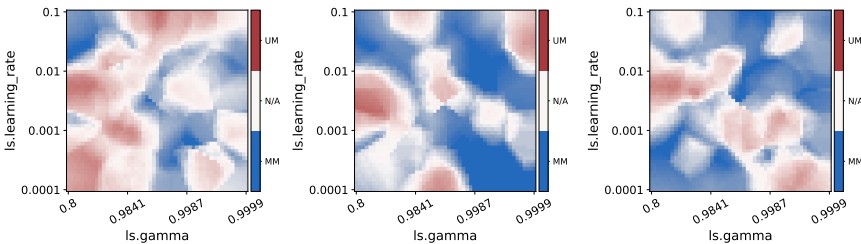

Figure 6: Discretized modality plots for learning rate and discount factor for DQN across phases

DQN, 60.94% configurations for SAC, and 80.47% for PPO. Although there are some configurations in this area that are classified as unimodal, their return distributions are otherwise not that optimal with their IQMs being dominated by other configurations that are not classified as unimodal. We additionally see that the unimodal configurations are almost double for DQN as compared to SAC and PPO, which correlates with the more complicated optimization problem of Hopper and BipedalWalker as compared to CartPole. While further analyses are necessary to ablate the various factors that impact modality, these observations contradict previous observations on benign landscapes of static algorithm configuration and AutoML (Pushak and Hoos, 2018, 2022; Schneider et al., 2022).

## 6 Conclusion

We presented a pipeline for data collection, landscape modeling, and landscape analysis, introducing hyperparameter landscape analysis in the domain of AutoRL. Our multiphase approach gathers performance data at distinct points of training, which we subsequently used to build different landscape models. We further outlined how the landscapes can be analyzed to gather insights about hyperparameter optimization in the context of AutoRL.

We applied the discussed approach to the training of DQN on Cartpole, PPO on Bipedalwalker, and SAC on Hopper, where we found drastic changes in the hyperparameter landscape over time, suggesting that the use of dynamic configurations in RL may be well-motivated. We additionally showed that the stability of configurations is rather unpredictable depending on a context that is informed jointly by the learning dynamics of the algorithm and the exploration problem. However,

| Category | DQN | | | SAC | | | | PPO | | |
|---|---|---|---|---|---|---|---|---|---|---|
| | Phase 1 | Phase 2 | Phase 3 | Phase 1 | Phase 2 | Phase 3 | Phase 4 | Phase 1 | Phase 2 | Phase 3 |
| % Unimodal | 19.53 | 13.28 | 15.62 | 19.53 | 09.37 | 06.25 | 07.81 | 12.5 | 10.94 | 8.59 |
| % Multimodal | 40.63 | 60.94 | 60.16 | 49.22 | 53.90 | 57.81 | 60.94 | 67.18 | 60.16 | 80.47 |
| % Uncategorized | 39.84 | 22.66 | 22.66 | 28.90 | 30.47 | 24.22 | 27.34 | 20.31 | 28.90 | 10.94 |

Table 2: Percentages of configurations assigned to each of the classes from the modality analysis.

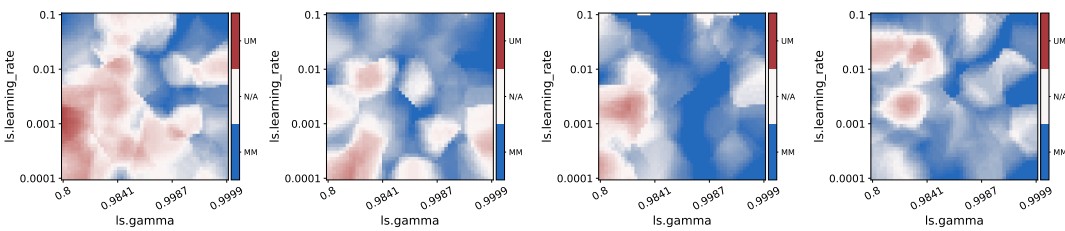

Figure 7: Discretized modality plots for learning rate and discount factor for SAC across phases

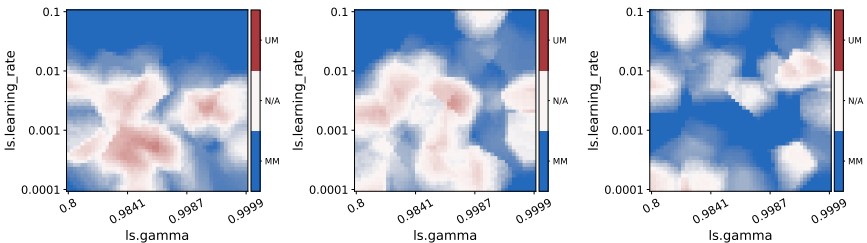

Figure 8: Discretized modality plots for learning rate and discount factor for PPO across phases

comparisons between algorithms could be made based on this fact to inform algorithm selection and algorithm creation. Consequently, we hypothesize that current multi-fidelity approaches using learning curves of RL training cannot factor in the dynamic hyperparameter landscape and thus might not be optimal for RL. Finally, we examined the modality of the return distributions and determined that only a small fraction ends up being unimodal, in contrast to the recent observations of benign landscapes in AutoML and algorithm configuration (Pushak and Hoos, 2020, 2022; Schneider et al., 2022). This shows that the dynamic configuration of RL agents poses a much harder problem than classical static AutoML addresses so far and calls for new and specialized AutoRL methods.

## 7  Limitations and Future Work

Our method of creating HP landscapes opens up a gateway to more principled analyses of HP configurations in AutoRL, which we consider highly important for deriving HP schedules that are more informed by the learning dynamics of the algorithm and the nature of the optimization problem. Currently, our method works only for continuous HPs and on a limited number of phases. A natural extension of our approach is incorporating other types of HPs in RL, albeit with the appropriate distance between categorial HPs being a central question. We see the usage of quasi-distance via the performance space as a potential direction for such work. Another major extension is to capture the change of the landscape in a function from which we can derive dynamic optimizers for RL.

**Broader Impact Statement**: After careful reflection, the authors have determined that this work presents no notable negative impacts on society or the environment.

## Acknowledgements

Aditya Mohan, Carolin Benjamins, and Marius Lindauer acknowledge funding by the European Union (ERC, "ixAutoML", grant no.101041029). Views and opinions expressed are those of the author(s) only and do not necessarily reflect those of the European Union or the European Research Council Executive Agency. Neither the European Union nor the granting authority can be held responsible for them.

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

## A  Full Plots

In the following sections, we present the full plots generated by our landscape data, which include **Combined Landscape plots**, **IGPR Maps**, **IGPR ICE curves**, **ILM Maps**, **ILM ICE curves**, **Modality plots**.

The data has been presented across phases, with the initial phase at the bottom of the page and the final page at the top. For the Maps, the upper triangles represent local maxima, while the lower ones represent local minima. Additionally, for SAC since the variation between the first and the second phases was low, we presented results second phase onwards.

## B  Model Fit

In order to gain insight into how well the IGPR model fits the data, we calculate the mean squared error and the mean absolute error on a 5-fold cross-validation procedure. For fitting the model the performance data is normalized to the range $[0, 1]$. In Table 3 we see that the IGPR model is able to fit the performance data of DQN and PPO quite well whereas it shows higher errors for SAC. Compared with ILM, Table 4, we see very similar fits.

| Phase | SAC Mean squared error | DQN Mean squared error | PPO Mean squared error |
|---|---|---|---|
| 1 | $0.5160 \pm 0.3325$ | $0.0055 \pm 0.0012$ | $0.0027 \pm 0.0007$ |
| 2 | $2.5027 \pm 0.8373$ | $0.0564 \pm 0.0229$ | $0.0078 \pm 0.0039$ |
| 3 | $2.6486 \pm 0.4524$ | $0.0648 \pm 0.0329$ | $0.0212 \pm 0.0050$ |
| 4 | $2.8879 \pm 0.2562$ | NaN | NaN |

Table 3: IGPR Model Fit

| Phase | SAC Mean squared error | DQN Mean squared error | PPO Mean squared error |
|---|---|---|---|
| 1 | $0.4764 \pm 0.3680$ | $0.0044 \pm 0.0008$ | $0.0028 \pm 0.0006$ |
| 2 | $2.3295 \pm 0.6123$ | $0.0466 \pm 0.0187$ | $0.0076 \pm 0.0034$ |
| 3 | $2.0241 \pm 0.4517$ | $0.0587 \pm 0.0281$ | $0.0211 \pm 0.0059$ |
| 4 | $2.1405 \pm 0.3728$ | NaN | NaN |

Table 4: ILM Model Fit

# C  DQN IGPR Maps

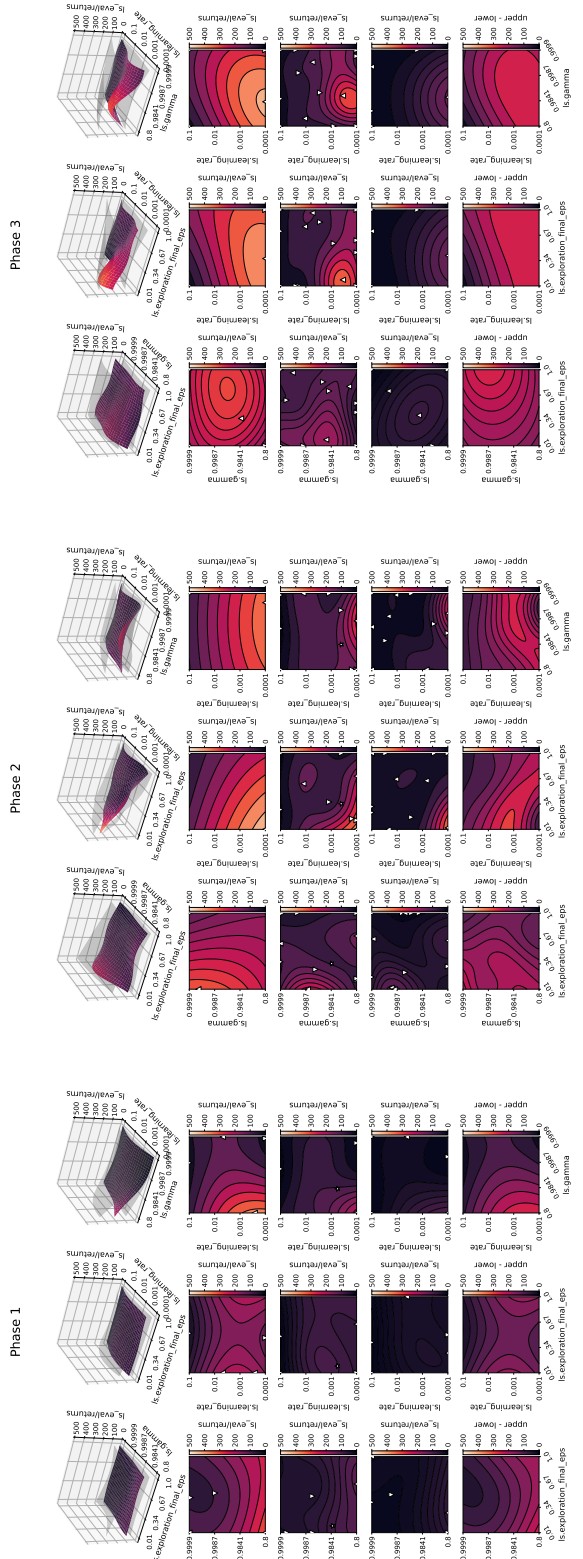

# D DQN IGPR ICE curves

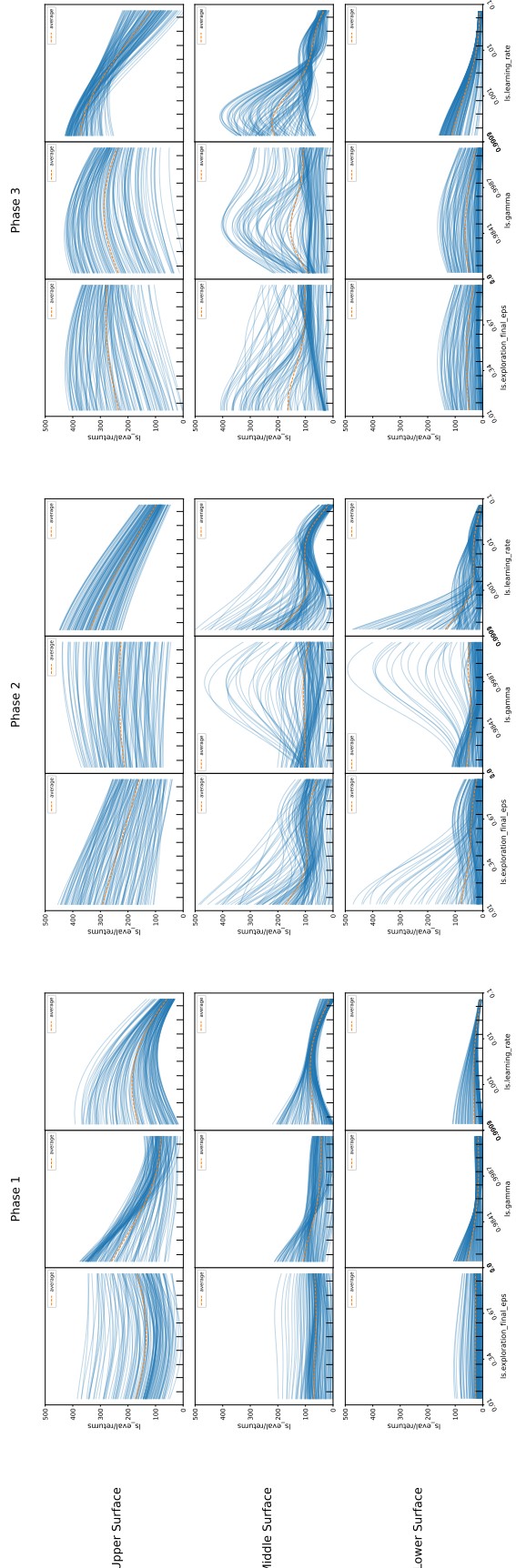

# E DQN ILM Maps

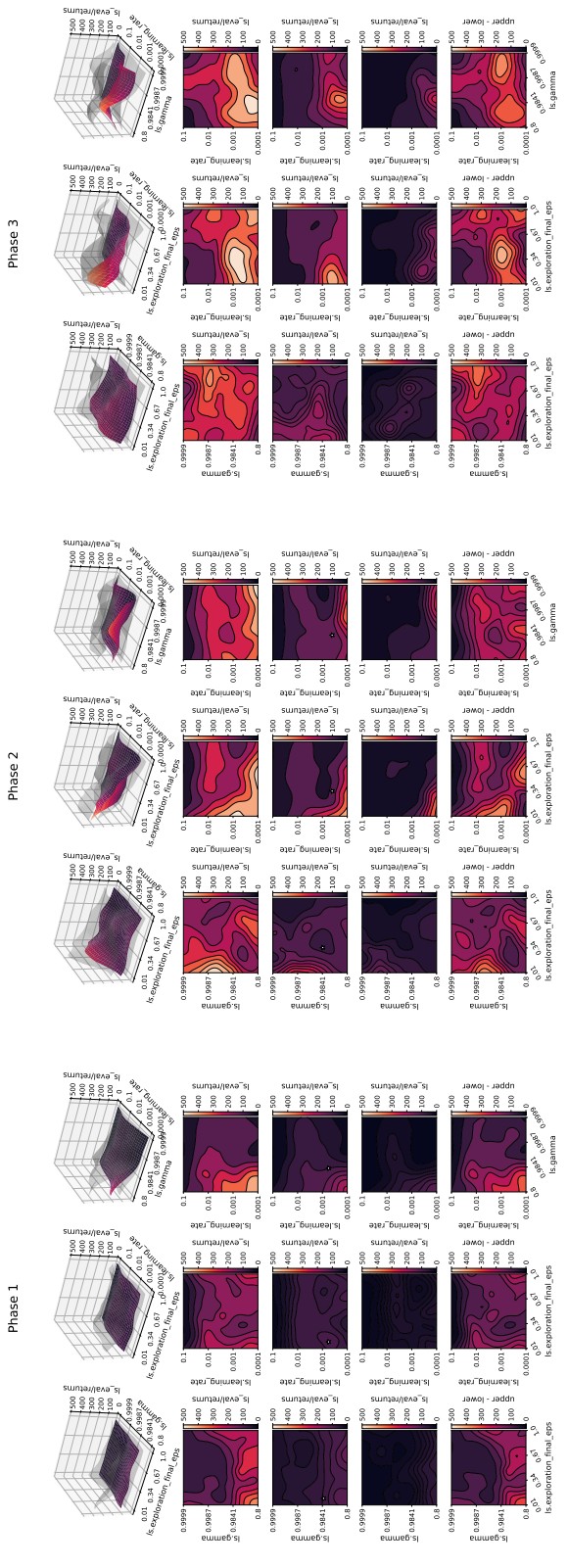

# F DQN ILM ICE curves

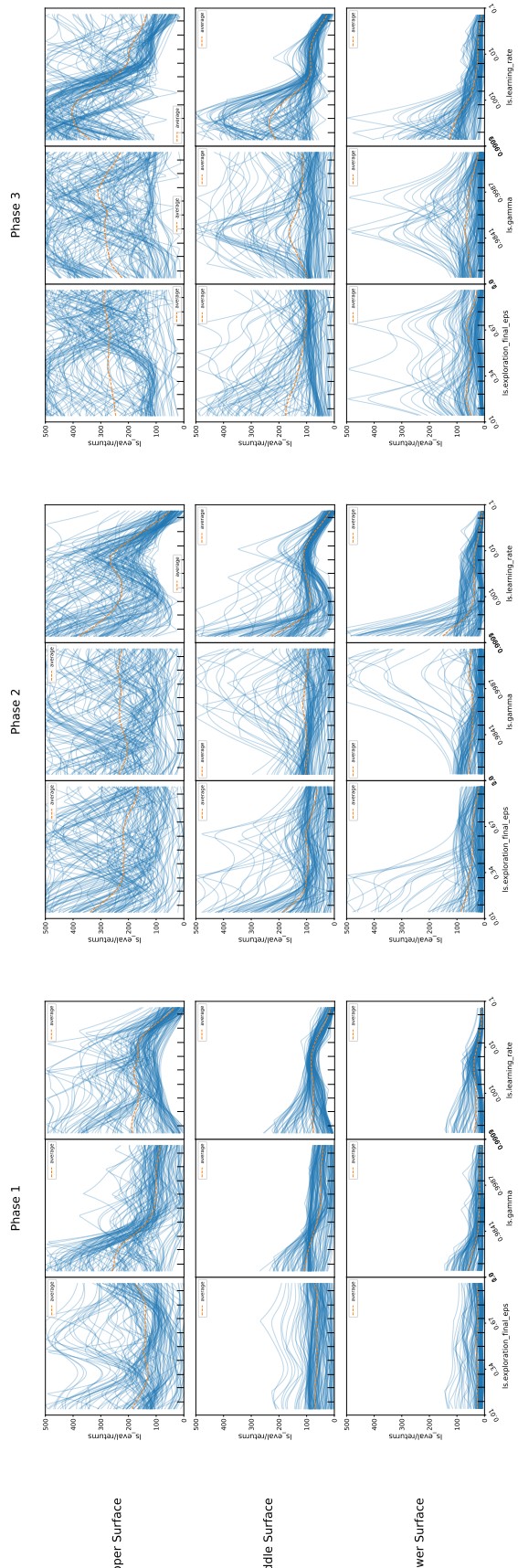

# G  DQN Modalities

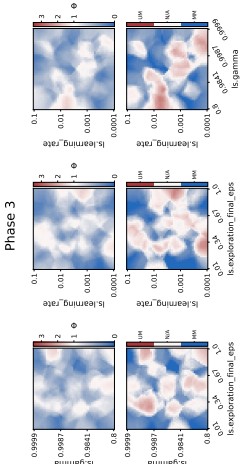

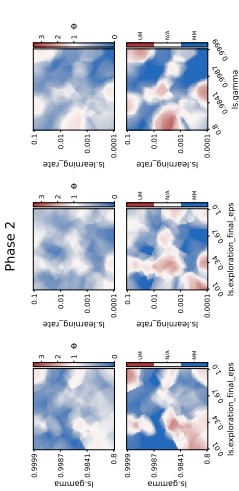

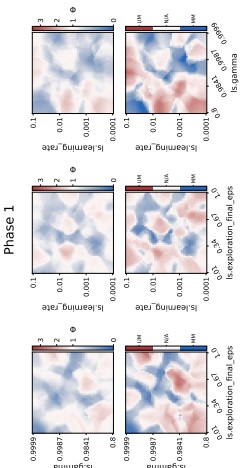

# H SAC IGPR Maps

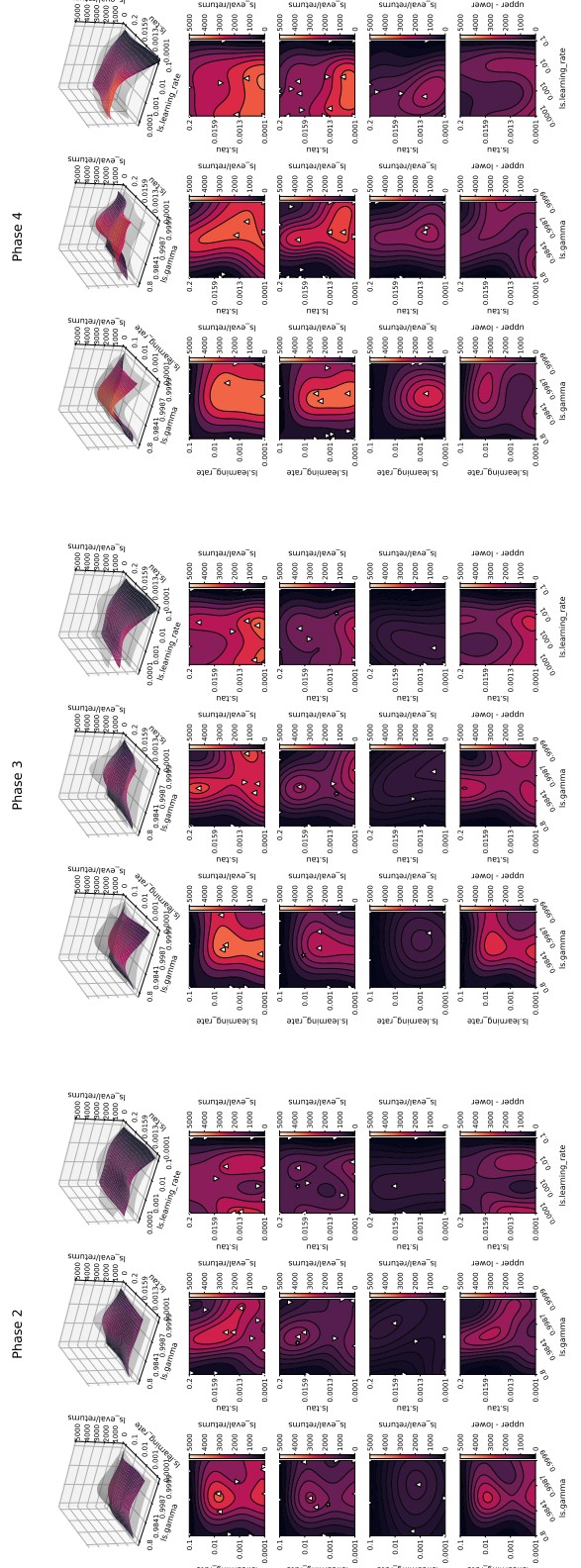

# I SAC IGPR ICE curves

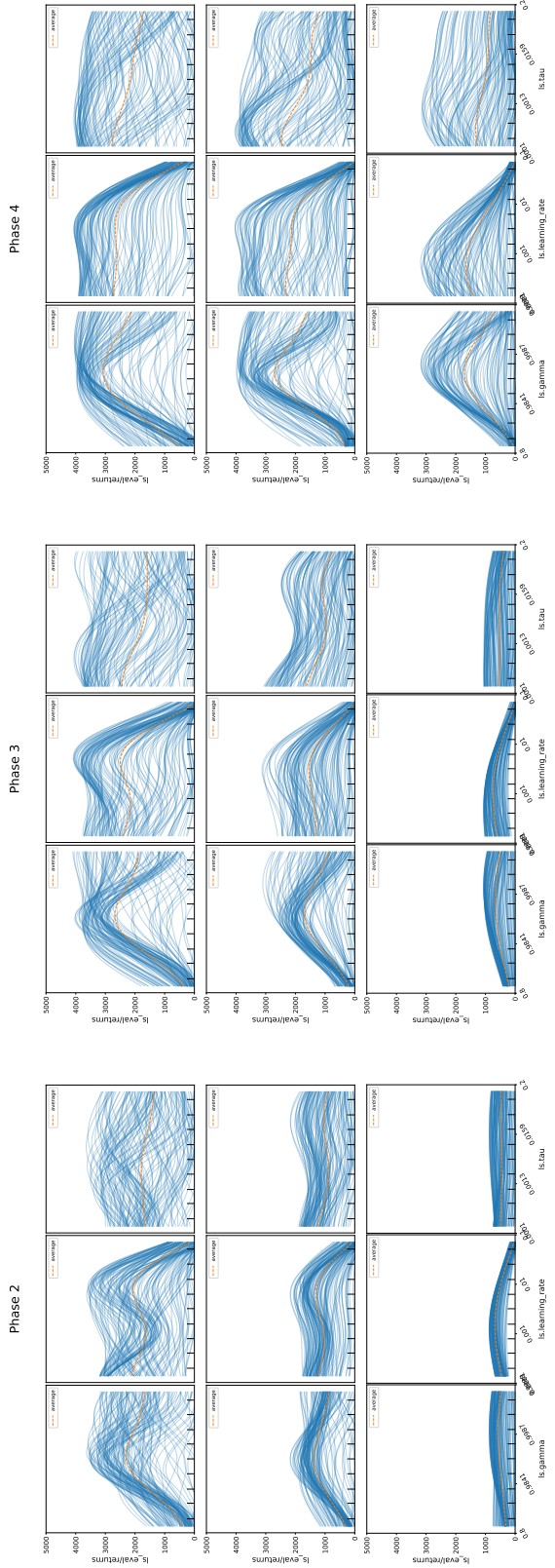

# J SAC ILM Maps

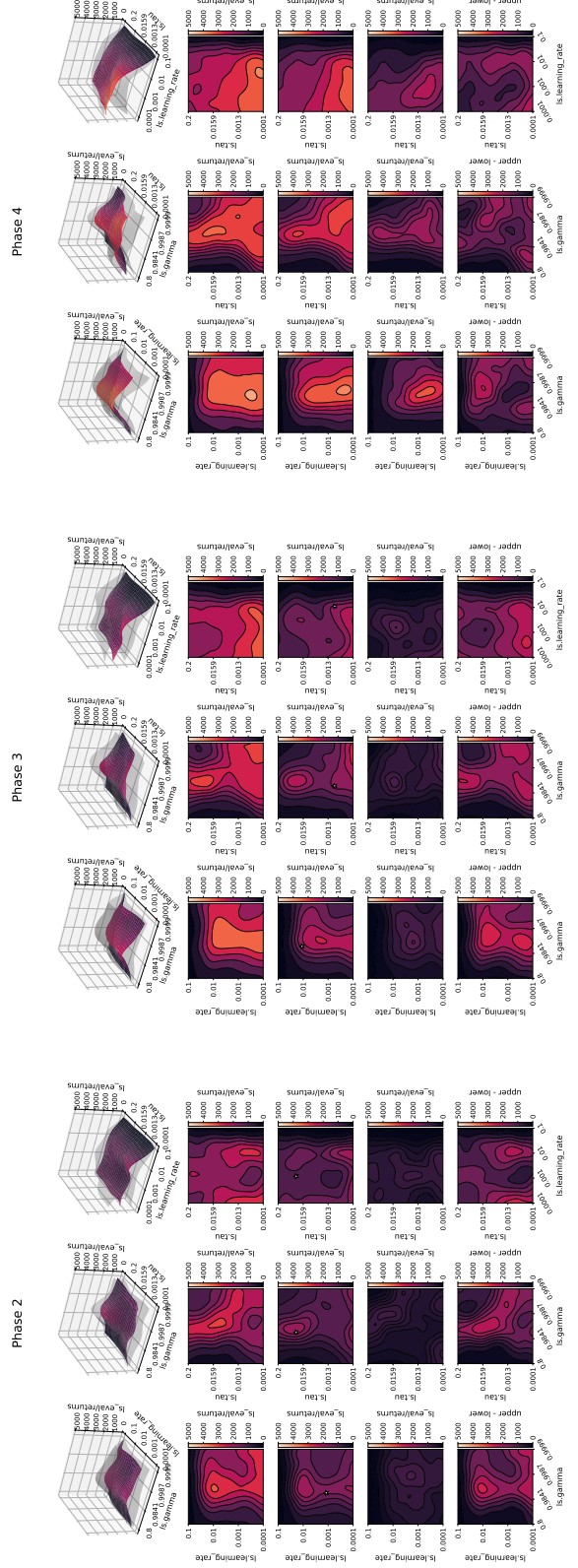

# K SAC ILM ICE curves

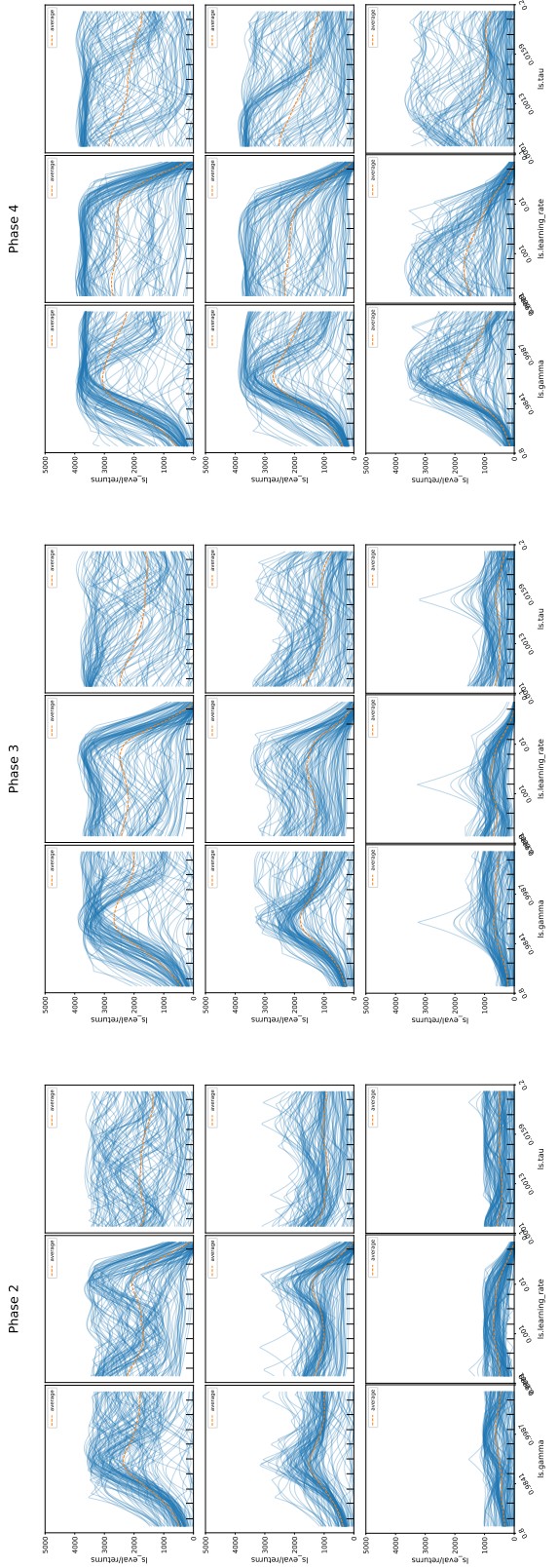

# L  SAC Modalities

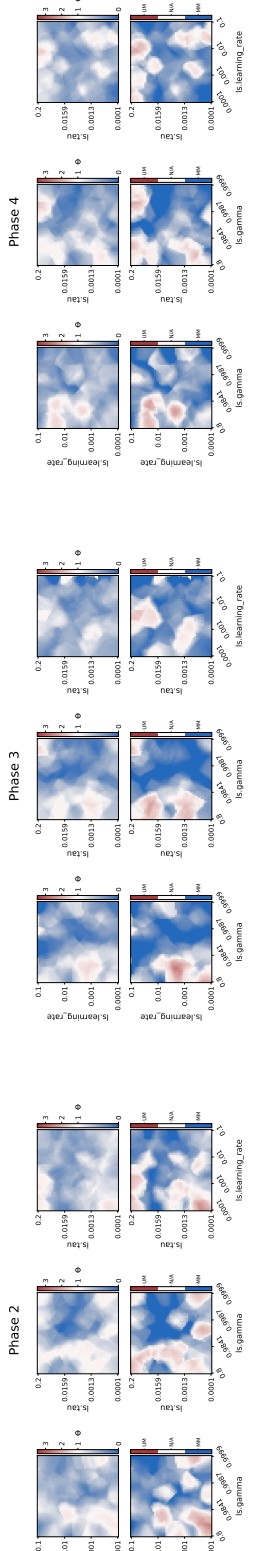

# M  PPO Results

We also evaluated PPO (Schulman et al., 2017) with three phases on Bipedal-Walker-v2 (Brockman et al., 2016b). We vary the hyperparameters learning rate, discount factor (gamma) and the generalized advantage estimate factor (gae_lambda) with the ranges specified in Table 5. From the IGPR approximations of the mean surface in Figure 9 we see that there is one region with high performance whose location also change over the phases. This implies that PPO is less robust to HP decisions in general. In addition, if we regard multi-fidelity optimization (Li et al., 2018) lower fidelities might not be good proxies for the target fidelity. Similar to DQN PPO has more multimodal configurations, with a high number of 80% for the last phase, see Table 6 underlining the volatile learning behavior of PPO. We attribute this partially to the learning dynamics of PPO (Lyle et al., 2022). This corroborates with the ICE curves in the final phase for all three hyperparameters in Figure 10. Across all phases, for the learning rate we see the same tendency of performance but not so for the discount factor and gae_lambda.

| PPO | | |
|---|---|---|
| HP | Range | Scale |
| Learning rate $\alpha$ | $[1e^{-4}, 0.1]$ | Log |
| Discount factor $\gamma$ | $[0.8, 0.9999]$ | Log |
| Generalized advantage estimate $\lambda_{gae}$ | $[0.8, 0.9999]$ | Log |

Table 5: Hyperparameters considered as part of $\lambda$ in PPO

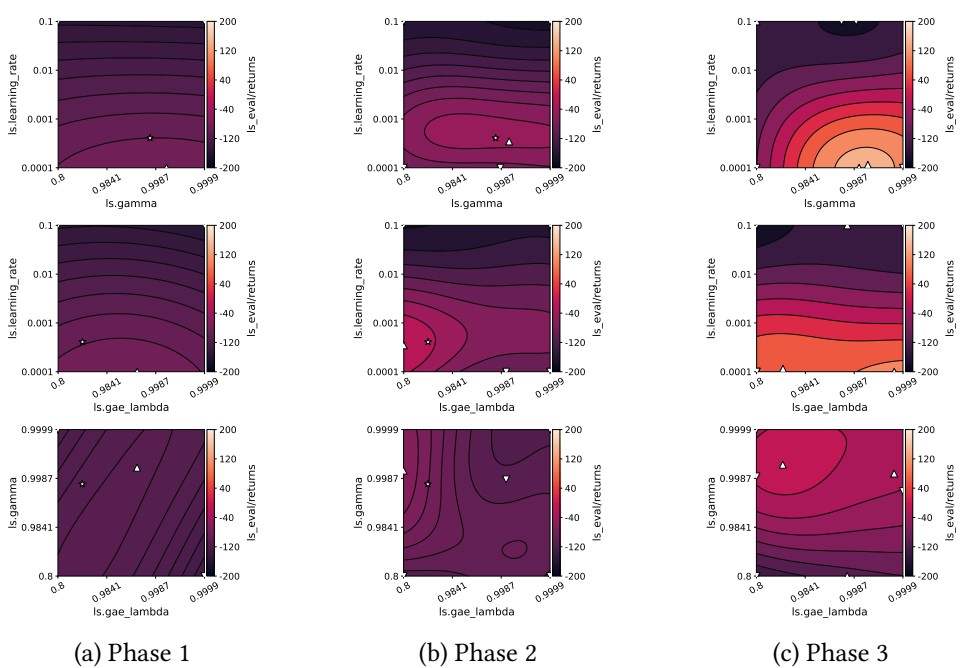

(a) Phase 1  (b) Phase 2  (c) Phase 3

Figure 9: IGPR plots of the middle surfaces for learning rate and discount factor for PPO across three phases of the RL training process.

| Category | PPO | | |
|---|---|---|---|
| | Phase 1 | Phase 2 | Phase 3 |
| % Unimodal | 12.5 | 10.94 | 8.59 |
| % Multimodal | 67.18 | 60.16 | 80.47 |
| % Uncategorized | 20.31 | 28.90 | 10.94 |

Table 6: Percentages of configurations assigned to each of the classes from the modality analysis

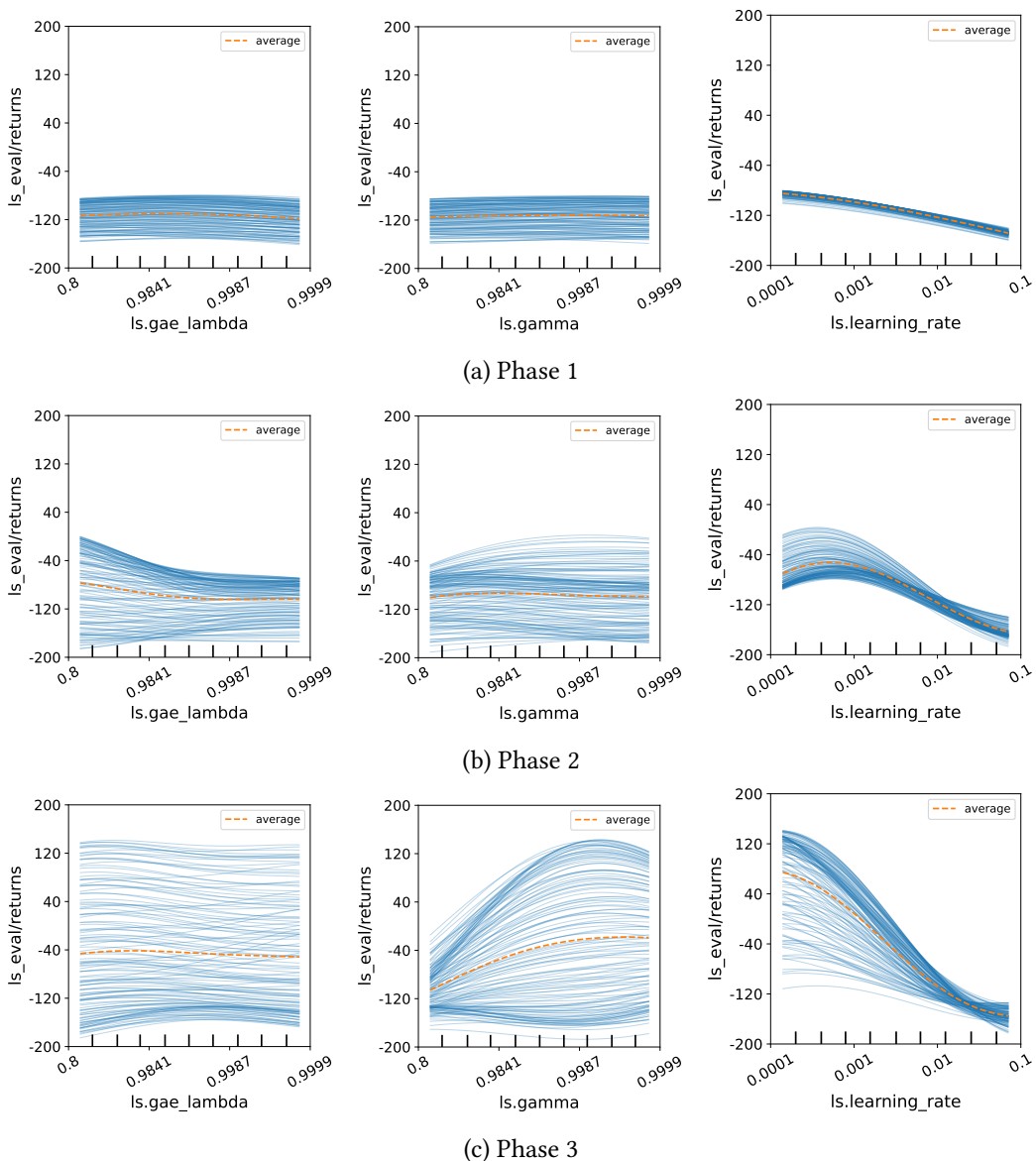

Figure 10: ICE curves for the middle surfaces for PPO across three phases of the RL training process (gae_lambda, gamma and learning rate)

