# OpenReview forum: "AutoRL Hyperparameter Landscapes"
_automl.cc/AutoML/2023/Conference — AutoML 2023 MainTrack_

### Official Review · Reviewer_7fjf · 2023-04-08

**Potential Impact On The Field Of Automl Rating:** 3
**Technical Quality And Correctness Rating:** 3
**Clarity:** The paper is well-written and well-or…
**Clarity Rating:** 3

**Summary Of Contributions:**

The article tries to improve the impact of hyperparameters on Reinforcement Learning (RL) via Automated RL (AutoRL).
The article proposes an approach to build and analyze hyperparameter landscapes at multiple points in time throughout training, showing strong variations over time for different RL algorithms in various environments. This supports the theory that hyperparameters should be dynamically adjusted during training and provides the potential for more insights on AutoRL problems through landscape analyses.

**Actions Required To Increase Overall Recommendation:**

I am willing to adjust my score based on the authors' responses if they could properly address my questions above.

**Overall Review:**

In general, I think this is an interesting paper, with clear methods and concrete results.
I hope the authors could address my questions, especially how the RL community would concretely benefit from this landscape analysis.

**Potential Impact On The Field Of Automl:**

The proposed landscape analysis could potentially help the RL community to analyze the hyperparameters and find better configurations.

**Review Confidence:**

4: You are confident in your assessment, but not absolutely certain. It is unlikely, but not impossible, that you did not understand some parts of the submission or that you are unfamiliar with some pieces of related work.

**Review Rating:**

7: Weak Accept: Technically sound paper with moderate-to-high impact and strong evaluation, with perhaps some minor flaws.

**Review Summary:**

Based on my understanding and questions, I would rate weak acceptance.

**Technical Quality And Correctness:**

I check the method section and the techniques seem sound.
Some of my questions are explained below.

1. Why do you need to create different phases? How one could determine the most suitable number of phases?

2. Could the authors elaborate in detail on how one could leverage your method (create and analyze landscapes) to their RL research and applications? Of course, it helps the visualization, but what’s the benefit to better performance or principled understanding?

3. How accurate is the IGPR landscape? Some possible indicators are: 1) the best configuration estimated by IGPR from the plot can outperform the best one in the collected data; 2) from the collected data, create a train-validation set split, use the train set to build the IGPR plot, and calculate the correlation b/w IGPR estimations and the validation set.

4. Is this landscape analysis a general method or dedicated to RL, and why? It seems to me that this method is agnostic to the learning task.

---

> ### Author Response · Authors · 2023-04-28
> **Initial Response**
>
> Thank you very much for your review. Please find the answers to your questions below.
>
> ## Phases
>
> Our motivation for inspecting the hyperparameter landscape at different points during training stems from previous work showcasing the potential of dynamic algorithm configuration [1] as well as the empirical performance of dynamic AutoRL methods like PBT and variants [2].
>
> RL is an especially non-stationary process as the learning data is generated by the also changing policy. Thus, in building the HP landscape, we wanted to not just cover the space of hyperparameters effectively, but also do it tractably and in a way that is intimately tied to the data generation process. Consequently, we divided the training into phases that allowed us to characterize the landscape at different training points, and we generated configurations from a policy performing best in the end. An ideal number of phases as well as the duration is indeed an interesting question and requires further investigation in our opinion, and we plan to look into this further in follow-up work.
>
> ## Leveraging the landscapes for RL research
>
> Our method is a principled way to analyze the impact of HP configurations on the RL pipeline, opening up opportunities to better understand the interplay between exploration, credit assignment, and HP configurations. We believe it can be utilized by practitioners in the future to perform more in-depth studies on how hyperparameters impact SOTA methods in their optimization procedure, or how HPs impact monolithic and non-monolithic exploration strategies. Further, we see a big potential for using these trends to design better optimizers for RL. These are also the directions that we intend to follow up on.
>
> ## Accuracy of Models
>
> We will add a paragraph on the accuracy of the selected models until the end of the rebuttal.
>
> ## Applicability
>
> The landscape analysis, as proposed in the current state, is applicable to only RL pipelines. However, similar analyses have previously been performed for supervised learning settings [3], and have traditionally been a part of the convex and evolutionary optimization community [4]. Along this line, one intersection of immediate interest is the evolutionary [5] and Quality Diversity subcommunities [6] within RL since they tackle sequential decision-making through parallel optimization methods.
>
> [1] Adriaensen, Steven, et al. "Automated dynamic algorithm configuration." Journal of Artificial Intelligence Research 75 (2022): 1633-1699.
> [2] Jaderberg, Max, et al. "Population based training of neural networks." arXiv preprint arXiv:1711.09846 (2017).
> [3] Pushak, Yasha, and Holger Hoos. "AutoML Loss Landscapes." ACM Transactions on Evolutionary Learning 2.3 (2022): 1-30.
> [4] Malan, Katherine Mary. "A survey of advances in landscape analysis for optimisation." Algorithms 14.2 (2021): 40.
> [5] Salimans, Tim, et al. "Evolution strategies as a scalable alternative to reinforcement learning." arXiv preprint arXiv:1703.03864 (2017).
> [6] Parker-Holder, Jack, et al. "Effective diversity in population based reinforcement learning." Advances in Neural Information Processing Systems 33 (2020): 18050-18062.

---

### Official Review · Reviewer_9o8e · 2023-04-11

**Potential Impact On The Field Of Automl Rating:** 2
**Technical Quality And Correctness Rating:** 3
**Clarity Rating:** 3

**Summary Of Contributions:**

This paper provides an empirical analysis of the hyperparameter landscapes for AutoRL. The authors collect a dataset using DQN and SAC on two RL environments that comprises trajectories collected from various hyperparameter and seed combinations, and divide the RL training into phases so as to study the temporal variation of the landscapes. From these empirical results, the authors conclude that the hyperparameter landscapes vary strongly over time, which is consistent with the assumptions that various AutoRL algorithms already depend on, but such phenomenon also questions the applicability of the multi-fidelity approaches in the context of RL. The finding that many hyperparameter landscapes are highly multi-modal also further hints the complexity of the AutoRL problems.

**Actions Required To Increase Overall Recommendation:**

Please answer my questions and address my comments especially in the `Impacts` section.

**Clarity:**

The paper is largely well-written and clear. Some minor typos & notational issues:

- line 188: initial -> initialize?

- Eq 3: I assume the calligraphic J means the space of possible objectives, but this symbol has not been defined.

**Overall Review:**

## Strengths
- The paper addresses an important problem in AutoRL, and fills the gap of providing empirical support for an assumption widely used in the literature, and as such it is definitely relevant to the AutoRL/ML audience for this conference.
- The methods presented are technically sound, and the findings are in general well-supported and could be useful for future AutoRL works.

## Weaknesses
- On the flip side of Strengths point 1, the paper largely stops at only confirming the assumptions held by the community but did not go further. As I mentioned in `Potential Impact`, I do expect an analysis paper like this to provide further insights especially on their implications on the existing AutoRL practices and suggestions for future AutoRL directions. I feel this is particularly true given that the analysis techniques used in this paper are not novel in their own right, and thus the resulting analyses and insights should form the bulk of this paper's contributions.

**Potential Impact On The Field Of Automl:**

I think the paper is relevant for the AutoRL community as it fills the gap by providing empirical evidence to commonly held assumptions, which is definitely nice. However, while I am indeed not aware of previous works that have similarly methodically studied the assumptions that "dynamic scheduling of hyperparameters in AutoRL is important" and "AutoRL presents a more challenging problem than general AutoML" like this paper, the success of numerous previous AutoRL algorithms that exploit these assumptions provide implicit evidence. In this sense, this paper confirms something already widely believed to be true, but largely *only* achieves that.

Given that the paper largely relies on existing analysis techniques adapted in the context of AutoRL, I do expect more and deeper analyses especially w.r.t. the state of the art for analysis paper like this. While currently some statements such as unsuitability of multi-fidelity approach are scattered around the paper, discussion in this aspect, which I think is important, is still largely underdeveloped in my view.

Some examples that I believe this paper may show larger potential could be: 1) in light of the landscape analysis as a "ground-truth" reference, a comparative study on whether existing AutoRL algorithms sufficiently explore the landscapes and genuinely adapt to the temporal shifts as intended? If that's not the case, what are the suggested ways to address these problems? 2) what are the implications of the prevalence of multi-modality, especially in relation to the existing AutoRL approaches? Do they handle these scenarios well?


**Reproducibility (Optional):**

I have no concerns on the reproducibility of this work.

**Review Confidence:**

4: You are confident in your assessment, but not absolutely certain. It is unlikely, but not impossible, that you did not understand some parts of the submission or that you are unfamiliar with some pieces of related work.

**Review Rating:**

6: Borderline Leaning Accept: Technically sound paper where reasons to accept outweigh reasons to reject. Please use sparingly.

**Review Summary:**

My initial recommendation is a borderline reject: while the paper shows promise and is largely technically sound, the suggested revisions may be material, and thus the paper would benefit from another round of reviewing.

I will reconsider the score based on the author feedback and the other reviews.

**Technical Quality And Correctness:**

I think the paper is largely technically sound. Some questions and comments below:

1. How do you determine the phases? It seems that currently the authors only use a regular interval (e.g. 50000 timesteps) to segment the training trajectory, but the training dynamics might vary greatly initially vs towards the end (where reward plateaus). I'm wondering something better could be done here.

2.  I am a bit concerned that the results are only collected on two rather small-scale environments, even though the authors tried to justify their decision in Line 236. However, I do think the main findings of the paper hold nonetheless.

3. The authors propose using GP to model the surface, and I'm wondering have the authors checked its modelling quality, by, for example, cross validation? I think the idea of using a surrogate for interpolation is a bit similar to NAS-Bench-301, where the authors there conduct thorough analysis on how well their predictor actually predicts. I think a similar analysis would also be beneficial here.

---

> ### Author Response · Authors · 2023-04-28
> **Initial Response**
>
> Thank you very much for your review. In addition to the points addressed in the general response (motivation, additional experiments and sampling) we would like to address your points in detail.
>
> ## Future Work
>
> We absolutely agree that linking the dynamic hyperparameter landscapes to different AutoRL methods and thus analyzing the adaptability capabilities is the next step for future work.
> Especially with respect to the prevalence of multi-modality, which is also highly relevant for multi-fidelity methods. In multi-fidelity, we need to promote good hyperparameter configurations with only a very limited number of evaluations which probably exhibit major multi-modality. This is a general challenge in AutoML and far from being solved.
>
> ## Determination of Phases
>
> As a first step we chose regular intervals as you mentioned. We agree that very probably the learning dynamics vary throughout the training and probing at those points in time might yield more insightful landscapes. Following this thought, this would mean determining the learning dynamics in the static case. However, we cannot guarantee the same dynamics for our dynamic case where we change hyperparameters during training.
>
> ## Surrogate Model Quality
>
> We thank you for raising this point which we deem very useful to select the model for representing the hyperparameter landscape. We will update the paper with information about the modeling quality until the end of the rebuttal.

---

> > ### Comment · Reviewer_9o8e · 2023-05-01
> > **Thank you for the response**
> >
> > I'd like to thank the authors for responding to my review, answering my questions and the additional experiments provided on the new environment. While my view of the paper improved somewhat, I still think this paper is borderline given my expectation that analysis paper like this provides deeper insights into the existing AutoRL algorithms, as I mentioned in the original review, a revision that I acknowledge is material and not necessarily feasible within the response period. I have updated the score, but I still think that the paper can be strengthened significantly from more time for a substantial revision and an additional round of review.

---

### Official Review · Reviewer_FvT5 · 2023-04-11

**Potential Impact On The Field Of Automl Rating:** 3
**Technical Quality And Correctness Rating:** 4
**Clarity Rating:** 4
**Actions Required To Increase Overall Recommendation:** Further diversification of the enviro…

**Summary Of Contributions:**

The paper focuses on the study of dynamic landscapes of hyperparameters during the training of RL algorithms. It focuses on DQN and SAC as representative DeepRL algorithms, and Cartpole and Hopper-v3 as the target environment for the study.
Paper introduces a pipeline for methodical collection of experiences with the hyperparameters during the "phases" of DeepRL training. The results are then visualized using IGPR and modality plots primarily in the paper and ILM, ICE plots shared in the appendix.
The paper highlights the change in optimality peaks across the phases and provides practical recommendation for HPO for DeepRL pipelines.

**Clarity:**

Paper is written with clarity. The initial sections provide concise yet clear background and the importance of work. Experimental setup and results were easy to follow.

**Overall Review:**

The paper provides useful insights into the landscape for HPO for DeepRL algorithms. Clearly defined experiments and shared results are valuable for the DeepRL community and will provide more concrete steps toward the realization of AutoRL.
The minor limitation of the study is in terms of the number and characteristics of the environments. It will be interesting to see if the results shared in the paper hold for more complicated and diverse environments (eg. discrete environment - Atari etc).

**Potential Impact On The Field Of Automl:**

Given the sensitivity of HPO for the DeepRL algorithms, the methodical study of its landscapes is a crucial step in creating a practical guide for HPO for DeepRL applications. The paper also provides support for the prior studies on the need for change in HP during the training phase.

**Review Confidence:**

4: You are confident in your assessment, but not absolutely certain. It is unlikely, but not impossible, that you did not understand some parts of the submission or that you are unfamiliar with some pieces of related work.

**Review Rating:**

9: Strong Accept: Technically flawless paper with major impact and strong evaluation, with no obvious flaws. Should be highlighted in the program.

**Review Summary:**

Great paper with fundamental (better understanding of the evolution of HPO landscape during training) and practical (a helpful guide to plan HPO for DeepRL algorithms) impact.

**Technical Quality And Correctness:**

The paper has explained all experimental setups and results with clarity. Results are explained well and the discussion is correlated with the results.

---

> ### Author Response · Authors · 2023-04-28
> **Initial Response**
>
> Thank you very much for your positive review of our work. We are elated that you see the benefit of the methodology we present and the potential it has to help the field of AutoRL. For your comment regarding environmental diversity, we have added the experiment of PPO on BipedalWalker to demonstrate the applicability of our approach to a different problem.
> We aim to scale this up to even more diverse environments in the future.

---

### Official Review · Reviewer_NcQo · 2023-04-13

**Potential Impact On The Field Of Automl Rating:** 2
**Technical Quality And Correctness Rating:** 3
**Clarity Rating:** 2

**Summary Of Contributions:**

This paper performs an empirical investigation of hyperparameter loss landscapes for two RL tasks (Cartpole and the Hopper environment from MuJoCo) using two algorithms (DQN and SAC). The authors focus on the dependence of each method/environment on hyperparameters at different points during training, motivated by prior empirical results showing the dynamically modifying hyperparameters during training can be beneficial compared to using static hyperparameters.

The authors visualized the hyperparameter loss landscapes at iterations 50k, 100k, and 150k for DQN, and at iterations 125k, 250k, 375k, and 500k for SAC. They found that the landscapes vary significantly at these points in training, supporting the notion of using hyperparameter schedules. They also plotted a measure of multimodality of the returns produced by any given hyperparameter configuration, across the hyperparameter space, and found that configurations become more multimodal as training progresses.

**Actions Required To Increase Overall Recommendation:**

I think that the writing and results need to be presented more clearly, and the diagrams (e.g., Figures 1 and 2) should be revised to improve clarity. I would encourage the authors to provide more analysis on larger-scale RL tasks and investigate which types of hyperparameter schedules are best for a given problem setting, to provide more practical takeaways.

**Clarity:**

The paper has several clarity issues, in the writing as well as the diagrams (e.g., Figures 1 and 2). Please see the Overall Review box for details.

**Overall Review:**

**Pros**

* The paper studies an interesting problem, trying to understand the impact of hyperparameters over the course of training for RL tasks. Insights into the effects of these hyperparameters may be useful to understand why practical approaches that dynamically update the hyperparameters work well.

* The results showing that the hyperparameter loss landscapes change substantially during training support the idea that hyperparameter schedules are important in RL.


**Cons**

* Although I appreciate that running a sweep over different hyperparameters is not cheap, the empirical evaluation is still fairly limited, only investigating two algorithms (DQN and SAC) on two tasks (Cartpole and Hopper). These tasks are small-scale, so it is not clear whether the hyperparameter loss surfaces for larger-scale tasks would follow a similar pattern (e.g., perhaps the trends would be different, or they could be more or less pronounced).

* The paper does not provide practical takeaways or actionable recommendations about which schedules to use in practical situations.

* Overall, the clarity of the presentation could be improved. Figure 1 is not very well-designed and does not add much to the paper. The reader can only guess that the symbols in the "data collection" box are related to the data collection procedure from Figure 2 after having read several pages further in the paper. Also, Figure 1 seems too generic to be useful, because the steps of 1) collecting data for a performance dataset, 2) using that data to construct hyperparameter landscapes; and 3) analyzing that landscape, are very broad, and not specific to RL; they could just as well be applied to arbitrary ML tasks. From Figure 1, it is not clear what the "search space" on the left-hand side refers to. In the caption of Figure 1, it is not clear what "modeled with landscape functions" mean?

* The organization of the writing is awkward. There is a lot of space devoted to explaining the basics of RL (Section 3.1), that could likely be summarized in one paragraph. At the same time, important details like the sampling procedure for hyperparameters are not expanded on clearly. The paper could use another pass over the writing to reduce wordiness and improve clarity.

* In Section 4, it states that "we imitate an optimal AutoRL optimizer that would always go for the best possible hyperparameter configuration schedule." The way this is described, it seems like a chicken-and-egg problem: how does one know what the best hyperparameter configuration schedule is without first knowing how the hyperparameter loss landscapes look under different conditions?

* Why are the hyperparameter configurations sampled using the complicated scrambled Sobol sampling strategy? As this paper only considers three hyperparameters for each algorithm, it should be possible to run a simple grid search, which would be more interpretable.

* Figure 2 is hard to parse, and not clearly described in the main text. In addition, the caption is completely uninformative. The font size is too small, and the arrows between blocks are not easy to interpret (owing to the lack of expanation in the caption). Why does it look like Step 5 in Figure 2 is determining the best configuration based on $\mathcal{D}\_{ls(1)}$? In the "Training and Evaluation" paragraph, it states that policies are selected based on their final performance, implying that in each stage the selection is based on $\mathcal{D}\_{\text{final}(i)}$, not $\mathcal{D}\_{ls(i)}$.

* It is not made clear what the "middle surfaces" are in Figure 3.

* What do the white triangles denote in Figures 3 and 4? Are these the hyperparameters that were evaluated? They appear to be unevenly distributed in the space, and do not seem representative enough to generate informative contours. Because there are few hyperparameters here, would a simple grid search be applicable? This would yield an even distribution of points across the range that is plotted, and I think may be better suited for such contour plots.

* Why are 3 timesteps used for DQN, while 4 are used for SAC? How were the timesteps chosen?

* I think that the last paragraph of Section 5.2 (starting at line 269) is unnecessary; it only makes vague statements.

* In Section 4.2, modality is not explained clearly enough. Does multimodality here refer to the distribution of performance values obtaiend when using the same hyperparameters with different random seeds? Also, the "folding test of unimodality" should be explained further here, as this is central to the quantitative results.

* The submission did not provide source code.


**Minor**

* L121: "Thus, An" --> "Thus, an"

* Eq. 3 can easily fit on one line.

* In Eq. 3, I don't think that $\mathcal{J}$ or $\Pi$ are defined anywhere.

* The first two sentences in Section 3.3 are redundant. In addition, the rest of Section 3.3 is wordy and seems overly drawn out.

* In Section 4.1, it is not stated what the $ls$ in $t_{ls}$ stands for.

* In Section 3.3, I think $\pi_Z$ needs to be defined (e.g., as the policy that results from applying algorithm $Z$).

* L49: "The goal of AutoRL is to solve to facilitate" --> "The goal of AutoRL is to facilitate"

* L21: "As the research" --> "As research"

* L171: "we consider s set" --> "we consider a set"

* L188: "then initial the configuration" --> should this be "then initialize the configuration"?

* L205: "of the Sobol' sampler" --> "of the Sobol sampler"

* L243: "train them on" --> "train them with"

* L246: Typo, "10000" should be "100,000"

* L287: "further analyses is" --> "further analyses are"

* L303: "factore" --> "factor"

* L305: "being in contrast" --> "in contrast"

**Potential Impact On The Field Of Automl:**

The paper does not make direct methodological contributions to the field of AutoML, but the idea of measuring the impact of hyperparameters at different points during training for RL tasks is useful. However, the investigation is not very complete here, as it does not yield insights regarding what the optimal schedules would be.

**Reproducibility (Optional):**

The submission did not provide source code, so it is not clear whether it is reproducible.

**Review Confidence:**

4: You are confident in your assessment, but not absolutely certain. It is unlikely, but not impossible, that you did not understand some parts of the submission or that you are unfamiliar with some pieces of related work.

**Review Rating:**

3: Reject: For instance, a paper with technical flaws, weak impact, and/or weak evaluation.

**Review Summary:**

Overall, this paper presents a fairly straightforward, visualization-centric, empirical investigation into hyperparameter loss landscapes for RL algorithms. The authors do not provide many takeaways from the results, other than that they support the idea of using hyperparameter schedules. There are no takeaways regarding what types of schedules would be useful, and what properties of the tasks would affect the optimal schedules. The experiments are limited, as they focus on two small-scale RL tasks, which may not be representative of larger, real-world problems. I do not think that the contribution of the paper is substantial enough to meet the bar for acceptance.

**Technical Quality And Correctness:**

I believe that the method is technically sound, and the results are likely correct. Please also see the detailed comments in the Overall Review box.

---

> ### Author Response · Authors · 2023-04-28
> **Initial Response**
>
> Thank you for your detailed review. We have addressed your points regarding the experiments and sampling strategy in our general comments. Please find our responses to your individual points below.
>
> ## Selected Environments
>
> To provide more evidence, we added further results in the results section on PPO for three phases on the Bipedal-Walker-v2 environment. See global answer for more details, too.
>
> Given the problem of compute expense in running multiple sweeps across different environments, our choice was based on the following factors:
>
> - Diversity in environmental dynamics
> - How well does an algorithm perform in the given environment
>
> Given that the selected algorithms (DQN, PPO, and SAC) cover three different kinds of objectives in the RL literature, we selected environments from both the classic control suite and MuJoCo to cover both simple and more nuanced dynamics. Given that we see similar trends, we expect that our results should generally hold. However, our method is agnostic to environments and agents, and thus, it can be transferred to any environment-agent combination.
>
>  ## Clarity
>
> Thank you for raising the point of clarity regarding the figures. We updated the figures for better accessibility. In more detail:
>
> - Figure 2: We redesigned Figure 2 to more intuitively show the data generation.
> - Figures 3 + 4: We denote the middle as mean surfaces for better clarity. The normal triangles represent local maxima, while the inverted ones, the local minima. The star represents the configuration selected for the next phase. We have added text in the caption to clarify this.
>
> We will fix the result figures’ font sizes until the end of the rebuttal.
>
> ### Figure 1
>
> We agree that the notion of the different phases to tell it apart from the standard HPO procedure, as you rightfully mentioned, was missing in the figure and in the description, which we now updated. However we believe that this figure summarizes our work and serves as a primer for the upcoming sections.
>
> ### Figure 2
>
> We have significantly simplified Figure 2 to focus on the core aspects of our data collection strategy. We believe it should be more intuitive now since it concretely demonstrates an exemplary case of 3 configurations for 3 phases.
>
> ## Basics of RL
>
> Our rationale for explaining the basics of RL in such detail was to address the AutoML community in general while being more self-contained in our writing. While these points are generally well understood in the subcommunities related to RL, we believe these points help an uninitiated reader to better understand our methodology.
>
> ## “We try to imitate an optimal AutoRL optimizer”
>
> We apologize for the misleading wording here and agree that it does indeed imply the chicken and the egg problem you mention. What we meant to say in that paragraph was that our approach to building the landscape involved focusing on the best configuration instead of all possible configurations, and we determine the quality of the configuration based on the final performance instead of the performance at that timestep. This would in fact correspond to a very well informed and expensive approach for an AutoRL optimizer. We have corrected this line to make this clearer.
>
> ## Source Code
>
> We would like to point you to line 247 which is where we provided the link to our source code. We have additionally added this link to the abstract for easier accessibility
>
> We have corrected all the minor errors pointed out by you and hope to have a clearer version.
>
> ## Modality
>
> We have added an intuitive explanation of the folding test of modality that hopefully elucidates the meaning of $\phi$

---

### Official Review · Reviewer_W8u8 · 2023-04-13

**Potential Impact On The Field Of Automl Rating:** 2
**Technical Quality And Correctness Rating:** 4
**Clarity Rating:** 3

**Summary Of Contributions:**

This paper showcases an analysis of Hyperparameter Optimization for Automated Reinforcement Learning. Within the analysis of this process, the authors showcased the need for dynamic configurations for hyperparameters in AutoRL. Using IGPR and discretized modality plots showcased the dynamic hyperparameter landscape for DQN and SAC.


**Actions Required To Increase Overall Recommendation:**

My main suggestion is to provide more information on these hyperparameter landscapes within this paper. Specifically by plotting more scenarios with different algorithms. The use of a small proof for the need for dynamic hyperparameter configurations is interesting, but not impactful enough to warrant acceptance.

Page 8, figure 6 description: Disretized -> Discretized

**Clarity:**

The paper clearly identified its experimental methods, reasoning, and results. I had some confusion reading the initial contributions - specifically with the part discussing how plotting these landscapes could help inspect them “for traits such as their general structure, configuration stability, and hyperparameter importance.” This appeared at first as if this is discussed further in the paper.

**Overall Review:**

Strengths:

The authors introduce a new pipeline to showcase hyperparameter landscapes for dynamic hyperparameters with discrete training steps, or more specifically for AutoRL problems. The paper also showcases definite proof for the use of dynamic hyperparameter configurations for AutoRL algorithms.

Weaknesses:

One weakness is the lack of analysis of the dynamic hyperparameter landscapes. The follow-on shown in the paper for this behavior was that this proves that dynamic hyperparameter landscapes should be prioritized, not how the hyperparameters change or how they can be predicted. This more important aspect to the field here is the characterization of the hyperparameter landscape. Discussing potential ways to analyze these landscapes would be beneficial. Another potential way to provide more information is to run more experiments to show different hyperparameter landscapes.
One other potential weakness in this paper is the use of Sobol sampling. Grid sampling can be non-optimal, but in a landscape analysis study, grid search can provide a good baseline to study the complexities of the landscape.


**Potential Impact On The Field Of Automl:**

The paper is currently limited in scope just to the proof that dynamic hyperparameter configurations are needed for AutoRL. The general field of AutoML could likely use this paper as a very broad citation for proof, but it is unlikely to provide substantial changes to the trajectory of current research.

**Review Confidence:**

3: You are fairly confident in your assessment. It is possible that you did not understand some parts of the submission or that you are unfamiliar with some pieces of related work.

**Review Rating:**

4: Weak Reject: For instance, a paper with minor technical flaws, limited impact, and/or weak evaluation.

**Review Summary:**

This paper conducted several experiments into the hyperparameter landscape of AutoRL algorithms in order to prove the need for dynamic HP configurations, and to identify a novel way to track and analyze these HP landscapes. While interesting, this paper lacks some substantial contribution.

**Technical Quality And Correctness:**

This paper had a solid experimental procedure, and well explained experimental design choices. It lacks some further investigation into the hyperparameter dynamism of AutoRL algorithms.

---

> ### Author Response · Authors · 2023-04-28
> **Initial Response**
>
> Thank you for your review. Please find answers to your questions below.
>
> ## More Scenarios
>
> We added PPO on Bipedal-Walker as a new scenario in our revision. With our selection of diverse RL algorithms, encompassing policy-gradient based, value-based and entropy-based methods, we believe we are able to showcase the general dynamicity of the hyperparameter landscapes.
>
> ## Analysis of Landscapes
>
> We agree that our approach opens up many possibilities for further studies and insights in understanding AutoRL and thus the design of better AutoRL approaches. We strongly believe that  already the analyses we provide in the paper are able to highlight some important points that can be corroborated by the literature. For example, the HP sensitivity of PPO can be very well validated with the multiple papers that talk about the importance of every design decision in PPO [1]. In contrast, we see a more stable landscape for SAC on a problem of similar complexity, signifying and corroborating the stability of entropy-based methods. With the data we collected for this study, the community can build upon it and for example further examine  the interplay between the learning dynamics, exploration, and credit assignment.
>
> [1] Engstrom, Logan, et al. "Implementation matters in deep policy gradients: A case study on ppo and trpo." arXiv preprint arXiv:2005.12729 (2020).
>
>
> ## Sampling on the Landscape
>
> We specifically choose Sobol sampling to use the best of both worlds, random sampling and grid sampling. Like grid search, Sobol sampling evenly covers the space and at the same time introduces more variations along one dimension. This is especially important in the case of low-effective dimensionality where we have unimportant hyperparameters, which we do not know beforehand [Bergstra & Bengio, 2012].
>
> ## Potential Impact
>
> Analyzing hyperparameter landscapes may allow for the prediction of optimized hyperparameter trajectories later on. This requires a sound foundation and the proposed work shows how the analysis of a landscape could be done. Due to the focus of this paper, predicting trajectories is not part of it, but will be considered for future work.

---

### Review · Reproducibility_Reviewer_Ux74 · 2023-04-26

**Completeness Of Code And Dataset Supplement Rating:** 4
**Usability And Ease Of Reproducibility Rating:** 3

**Actions Required To Increase The Reproducibility And Overall Recommendation:**

The readme should be updated to address the issues mentioned during the review.

If possible, the code should be updated to address the issue I encountered with incorrectly structured output files.

**Completeness Of Code And Dataset Supplement:**

The code and dataset supplement includes all of the files necessary for running the experiments.

In addition to the code needed to reproduce the results, the repository contains configurations for running shorter versions of the experiments described in that paper. The repository also includes the results data for the experiments described in the paper. Both of these are greatly appreciated, as they made reproducibility much easier: the shorter experiments allowed for testing the code under the limited time available for the review, while the result data allowed the reproduction of the visualizations included in the paper.

There is only one very minor issue: The readme and the conda configuration file reference certain files that are not included in the project. This was presumably a result of forgetting to update the conda configuration, or as a holdover from older experiments. However, the missing files are not necessary to reproduce the experiments mentioned in the submitted paper. More details on this will be mentioned in the "Usability And Ease Of Reproducibility" section.

**Overall Reproducibility Review:**

Positives

Overall, I find the included code to be of a generally high standard. The setup is relatively easy to install, with the conda environment handling most of the work. However, some parts of the installation have to be performed manually, and some, such as installing the mujoco library, are not listed in the readme.

In terms of executing the experiments, the provided command line utility makes this process very easy. However, the Readme often does not contain correct instructions, with missing parameters or instructions for running experiments that aren't part of the paper.

The visualization aspects are very easy to reproduce using the included data file that contains the results of the experiments.

The included source code is easy to read and reasonably well commented


Negatives

It is somewhat difficult to ascertain the complete reproducibility of the experiments due to the time and computational resources required to perform the full experiments described in the title.

Running a smaller version of the experiment finished without issue and with a file that appeared mostly error-free. However, some experiments contained errors in the result files that made the file incompatible with the code used for visualization. Other experiments were executed successfully and without errors.


In particular, some results contained lines such as
lgudgeq4,azure-sun-2837,0.9175495952967472,0.0004649660296171357,0.8009369617607445,"[{'learning_rate': {'type': 'Log', 'lower': 0.0001, 'upper': 0.1}}, {'gamma': {'type': 'FlippedLog', 'lower': 0.8, 'upper': 0.9999}}, {'exploration_final_eps': {'type': 'Float', 'lower': 0.01, 'upper': 1}}]",Sobol,CartPole-v1,50,500,500,0.95,3,20,10,10,20,MlpPolicy,64,256,100000,128,100,0.16,10,DQN,123,456,44,local,4,1000,1,dev,online,xxxx-xxxx,automl-reproduce,test-4,"[5, 10]",128,5,46,1,None,2023-04-26_16-06-01,1,,,,,,,,,,,,,,,,,,,,,,,,,,,,,,,,,,,,,,,,,,,

or

fivuarto,trim-galaxy-981,0.0016802768112879108,0.999852653331282,0.05332016470222627,"[{'learning_rate': {'type': 'Log', 'lower': 0.0001, 'upper': 0.1}}, {'gamma': {'type': 'FlippedLog', 'lower': 0.8, 'upper': 0.9999}}, {'tau': {'type': 'Log', 'lower': 0.0001, 'upper': 0.2}}]",Sobol,CartPole-v1,50,500,500,0.95,3,20,10,50,20,MlpPolicy,64,256,100000,128,100,0.16,10,DQN,123,456,44,local,4,1000,1,dev,online,xxxx-xxxx,automl-reproduce,test,"[500, 1000]",256,5,45,1,None,2023-04-25_13-11-31,196,,,,9.3,,0.0016802768112879108,,,,,,,,,,,,,,,,,,,,1682430850.600665,0.05332016470222627,,,,,,,9.3,,5.445664405822754,,,0,0,,,,,,0.999852653331282

With the sequences of commas presumably causing errors when running the visualization, as such sequences do not appear in the data provided in the repository.


The readme, as well as the configuration files, need a quick clean up to correct wrong filenames and incorrect command line instructions.

The included code contains some commented-out code and todos.



**Review Confidence:**

3: You are fairly confident in your assessment. It is possible that you did not understand some parts of the submission or that you are unfamiliar with some pieces of the code or data.

**Review Rating:**

8: Accept, all aspects of this are reproducible with minor effort.

**Review Summary:**

Overall, the code is relatively easy to setup and execute, due to the provided environments, configuration files, and a command line interface for running the experiments. The code itself also appears to be of good quality, with appropriate structure and comments. As such, I recommend acceptance.

The visualization experiments were easy to reproduce exactly by using the provided results files of the full experiment.

The data generation was harder to reproduce due to computational requirements and time constraints. Further, some smaller experiments produced errors, but this could be a result of the code not being designed to handle experiments of smaller sizes. However, some experiments were still executed successfully, and I was able to visualize the resulting data.


Other than the errors mentioned above, the drawbacks are very minor. Some filenames need to be checked, and the readme could be improved, as it currently contains errors, does not include some parameters that have to be used for the command line tool to work correctly, and fails to mention some libraries that have to be installed.


Overall, the code was easy to use, mostly reproducible to the extent allowed by the time constraints of the reproducibility review, and well structured and commented. Some errors will have to be fixed to make sure that the code does not produce corrupted results files (or that the visualization correctly handles such files), and the readme will have to be updated.

**Summary Of Necessary Code And Dataset Supplement:**

The code supplement contains the source files needed for the following tasks:

- Functions for analyzing the features of the hyperparameter landscape, such as its modality and concavity
- Implementations of the DQN and SAC agents modified so that they can be saved and loaded deterministically
- Implementations of the landscape models RBFI and GP
- General code for running experiments and integrating with wandb for tracking experiments
- Code for creating visualizations
- Various utilities, such as for downloading wandb data to a local file
- A command line tool for easily running the experiments and producing visualizations
- conda and pip configuration files for setting up the necessary code environment
- The configuration files for running different types of experiments. In addition to the experiments presented in the paper, the repository includes configurations for running shorter version of the experiments. I appreciate the inclusion of these experiments,as they allow for verification that the code and the programming environment are running correctly before running the more costly main experiments.


The code and data supplements were provided via an anonymous github repository https://anon-github.automl.cc/r/autorl_landscape-F04D/README.md, downloaded on 18:18, 25th April 2023.


**Usability And Ease Of Reproducibility:**

Strengths:

The code is generally very easy to setup and run, with some exceptions.

The code uses the Anaconda Python distribution and includes a preconfigured conda environment which automatically sets up most of the required dependencies. The included readme file provides easy-to-follow instructions for setting up the environment.

Some sections of the code require additional libraries which cannot be installed as a part of the conda environment, but this is understandable. The code is structured in a way where the majority of it can still be used without requiring these libraries, which is a plus.

Additionally, the code includes a command line tool for executing the experiments, which makes execution very straightforward.

The experiments also integrate with wandb for tracking experiment data. This integration is handled automatically through the code and the provided command line tool, and is easy to use even for users unfamiliar with the platform.

The random seeds used are included in the configuration files, which is necessary for reproducibility.

The code itself is well organized, readable, and appropriately commented, which should make potential modifications or extensions easy.

Weaknesses:

There is one major issue and a number of minor issues in terms of the reproducibility

The major issue is that it was hard for me to ascertain the reproducibility of the experiments due to large computing time required and the limited time available for review. Because of this, reproducibility experiments were performed using the "quick" configuration, which uses 500 and 1000 phases, rather than the 250k or 500k phases of the original setup. This experiment was repeated multiple time with a different number of the num_confs parameter, which determines the number of configurations evaluated, with settings from 2-256. On a local machine running an Intel i7-8550U processor running 4 cores at 4.00GHz each, the largest configuration took approximately 20 hours to complete.

This experiments executed successfully and without errors, and produced a result file that contained information in line with the full experiment. However, some results files contained lines withs errors, which made the file not work correctly when used for the visualization part of the experiments. Other experiments executed successfully and without any errors, and could be successfully visualized.

There are a couple of minor issues with the setup

1. The included conda environment file references an incorrect pip requirements.txt file.

2. The readme instructions are incorrect or not up to date in several places:

2.1 The command: "phases run combo=dqn_cartpole ls=sobol_2 slurm=local phases=100k num_confs=256 num_seeds=5 wandb.entity=entity_name wandb.project=project_name"  does not work, as the configuration files for the parameters phases=100k and ls=sobol_2 are not included in the repository. This command also does not correspond to the parameters used in the paper, which were :

num_confs = 128
phases = 150k or 500k
ls = dqn or sac

When using these parameters, the code executes without issue. The command included in the readme was likely a leftover from some previous experiments which used those parameters.

2.2 The command is also lacking the required parameter wandb.experiment_tag, which is necessary in order to run the code

2.3 The commands for visualization, such as phases ana maps --data=path/to/data.csv --model={rbf,triple-gp} are incorrect. The correct format was ana maps path/to/data.csv {rbf,triple-gp}, without the parameter names. I would also prefer for the instructions to include the --savefig parameter, which saves the generated figures to disk.

2.4 The Dataset Description section contains a reference to data/XXXX-11/dqn_cartpole_3_phases.csv. This file does not seem to be included. Instead, two other csv files are included, corresponding to the experiments in the paper, and the structure of the files described in the readme appears to be correct.

These issues were very easy to resolve, and they should still be fixed before the source code is uploaded to a public repository.

4. The library mujoco has to be installed separately since the mujoco-py library included in the requirements.txt only provides the python bindings, not the library itself.

5. The code files contain sections that were commented off, and some sections marked as TODO. This does not majorly impact readability but should be cleaned up before a public release.

6. The code appears to include functionality to execute both locally and on a slurm cluster. However, no instructions are provided for executing on a cluster. It is possible that the code for executing on a cluster only works on the specific cluster used by the authors. However, if the slurm functionality is general enough that it can be used by anyone, I would appreciate if instructions could be added to the README

---

> ### Author Response · Authors · 2023-05-02
> **Initial Response**
>
> Thank you for your very thorough reproducibility review! We appreciate your feedback and fixed the issues you mentioned.
>
> > The included conda environment file references an incorrect pip requirements.txt file.
>
> We looked into the issue here, and it seems that it came from the anonymization process. We have changed the usernames to be unanimous in the anonymous repository
>
> > The readme and the conda configuration file reference certain files that are not included in the project. This was presumably a result of forgetting to update the conda configuration, or as a holdover from older experiments. However, the missing files are not necessary to reproduce the experiments mentioned in the submitted paper. More details on this will be mentioned in the "Usability And Ease Of Reproducibility" section.
>
> > The command: "phases run combo=dqn_cartpole ls=sobol_2 slurm=local phases=100k num_confs=256 num_seeds=5 wandb.entity=entity_name wandb.project=project_name"  does not work, as the configuration files for the parameters phases=100k and ls=sobol_2 are not included in the repository. This command also does not correspond to the parameters used in the paper, which were :
> num_confs = 128
> phases = 150k or 500k
> ls = dqn or sac
> When using these parameters, the code executes without issue. The command included in the readme was likely a leftover from some previous experiments which used those parameters.
>
> > The command is also lacking the required parameter wandb.experiment_tag, which is necessary in order to run the code
>
> > The commands for visualization, such as phases ana maps --data=path/to/data.csv --model={rbf,triple-gp} are incorrect. The correct format was ana maps path/to/data.csv {rbf,triple-gp}, without the parameter names. I would also prefer for the instructions to include the --savefig parameter, which saves the generated figures to disk.
>
> We updated the README.md accordingly. Now the commands should be fine.
>
> > The Dataset Description section contains a reference to data/XXXX-11/dqn_cartpole_3_phases.csv. This file does not seem to be included. Instead, two other csv files are included, corresponding to the experiments in the paper, and the structure of the files described in the readme appears to be correct.
>
> We updated the file structure in order not to be affected by the code anonymization.
>
> > The library mujoco has to be installed separately since the mujoco-py library included in the requirements.txt only provides the Python bindings, not the library itself.
>
> Thank you for spotting this. We added a pointer in the README.md for this.
>
> > The code files contain sections that were commented off, and some sections marked as TODO. This does not majorly impact readability but should be cleaned up before a public release.
>
> We plan to do a code cleanup and clear all commented sections and TODOs.
>
> > The code appears to include functionality to execute both locally and on a slurm cluster. However, no instructions are provided for executing on a cluster. It is possible that the code for executing on a cluster only works on the specific cluster used by the authors. However, if the slurm functionality is general enough that it can be used by anyone, I would appreciate if instructions could be added to the README
>
> We added small instructions for a custom slurm setup. This should work on any slurm cluster.

---

### Author Response · Authors · 2023-04-28
**General Response**

We thank all the reviewers for their helpful feedback. We are glad that the reviewers consider our approach to be relevant to the AutoRL community by filling the gap through empirical evidence to commonly held assumptions (Reviewer 9o8e), and potentially instrumental in analyzing hyperparameters and finding better configurations (Reviewer 7fjf). We also appreciate the reviewers acknowledging the solidity of our experimental procedure (Reviewer W8u8), noticing the benefit of our methodical study of HP landscapes, and considering it a crucial step in creating a practical guide for HPO for DeepRL applications  (Reviewer FvT5).

We would like to reiterate that the rationale behind our paper was to show the importance of landscape analyses and their importance to the field of AutoRL. This study is a first step in that direction, and our hope is that with this, we can bring the attention of larger parts of the AutoML community, a community known for adding data-driven principles to the design of ML pipelines, to work on further properties of AutoRL. We believe our methodology can be built upon by future work to explore a more nuanced interplay between hyperparameters, exploration, and credit assignment in RL.

While we will address individual points as separate replies, we would like to address some general points in this comment.
## Additional Experiments
We additionally evaluated PPO for three phases on the Bipedal-Walker-v2 environment, which can be found in the results section. For this experiment, we varied the learning rate, discount factor, and the generalized advantage estimate factor (gae_lambda). The plots show that PPO is able to achieve higher returns in very specific hyperparameter regions as opposed to SAC which achieves a generally stable performance across a wide range. This adds to previous arguments about PPO being highly sensitive to code-level optimizations [1,2].

We believe that, along the same lines as DQN and SAC, these results demonstrate that our methodology adds principled analyses of RL algorithms that can be potentially applied to understand the nuances between dynamic hyperparameter changes and the learning dynamics of the algorithms that they impact.

Our general comments regarding multi-fidelity optimization still hold. However, we would additionally like to point out that these results, the generated meta-data and the approach itself  are meant as a gateway to more interesting analyses that can be possible using our methodology. Along this line, we believe work in understanding learning dynamics and exploration behavior of the algorithms is a promising and crucial future direction.

## Sampling Strategy

While grid search is potentially more interpretable, we don't know whether dimensions are correlated [3]. This creates issues since a simple grid would be inefficient in effectively covering the search space and potentially hampers the scalability of our approach.

Our rationale for choosing Sobol sequences, as a low-discrepancy sequence, was to exploit its pseudo-random nature and its ability to cover more values along one dimension of the search space, addressing the potential low effective dimensionality. Thus, we see Sobol sequences as a way to get best of both worlds of grid search and random search. We further highlight that Sobol sequences are a well-known strategy for initial designs in optimizers such as Bayesian Optimization.


[1] Engstrom, Logan, et al. "Implementation matters in deep policy gradients: A case study on ppo and trpo." arXiv preprint arXiv:2005.12729 (2020).
[2] Huang, Shengyi, et al. "The 37 implementation details of proximal policy optimization." The ICLR Blog Track 2023 (2022).
[3] Bergstra, James, and Yoshua Bengio. "Random search for hyper-parameter optimization." Journal of machine learning research 13.2 (2012).

## Changelog
As a result of the feedback received we made several updates to the paper:

- Inclusion of more experimental results: PPO on Bipedal Walker
- Updated Figure 1 and its description
- Redesigned Figure 2 (data collection) and added a more comprehensive caption

We marked the updated text with green color.